

# Measurement Characteristics of an airborne Microwave Temperature Profiler (MTP)

Mareike Kenntner[1], Andreas Fix[1], Matthias Jirousek[2], Franz Schreier[3], Jian Xu[3], Markus Rapp[1]

[1] Deutsches Zentrum für Luft- und Raumfahrt, Institut für Physik der Atmosphäre, Oberpfaffenhofen, Germany
[2] Deutsches Zentrum für Luft- und Raumfahrt, Institut für Hochfrequenztechnik, Oberpfaffenhofen, Germany
[3] Deutsches Zentrum für Luft- und Raumfahrt, Institut für Methodik der Fernerkundung, Oberpfaffenhofen, Germany

*Correspondence to*: Mareike Kenntner (Mareike.Kenntner@DLR.de)

**Abstract.** The Microwave Temperature Profiler (MTP), an airborne passive microwave radiometer, records radiances in order to estimate temperature profiles around flight altitude. From these data the state of the atmosphere can be derived and
important dynamical processes (e.g. gravity waves) assessed. DLR has acquired a copy of the MTP from NASA-JPL, which was designed as a wing-canister instrument and is deployed on the German research aircraft HALO.

For this instrument a thorough analysis of instrument characteristics has been made. This is necessary to correctly determine the accuracy and precision of MTP measurements, and crucial for a retrieval algorithm to derive vertical profiles of absolute atmospheric temperatures.

Using a laboratory set-up, the frequency response function and antenna diagram of the instrument was carefully characterised. A cold-chamber was used to simulate the changing in-flight conditions and to derive noise characteristics as well as reliable calibration parameters for brightness temperature calculations, which are compared to those calculated from campaign data. Furthermore, using the radiative transfer model Py4CAtS, the sensitivity to the atmospheric layers around flight altitude was investigated.

It was found that using the standard measurement settings, the DLR-MTP's vertical range of sensitivity is limited to 3km around flight altitude, but can be significantly increased by adjusting the standard measurement strategy, including slightly weaker oxygen absorption lines and a different set of viewing angles. Calibration parameters do clearly depend on the state of the instrument; using a built-in heated target for calibration may yield large errors in brightness temperatures, due to a misinterpretation of the measured absolute temperature.

With here presented corrections to the calibration parameter calculations, the measurement noise becomes the dominant source of uncertainty and it is possible to measure the atmospheric temperature around flight level with a precision of 0.38 K.

This is the first time such a thorough instrument characterisation of a MTP instrument is published. With the presented results, it is now possible to identify significant temperature fluctuation signals in MTP data and choose the best possible
measurement strategy fitting the purpose of the measurement campaign.


# 1 Introduction

Aircraft campaigns have long been used to study atmospheric composition and dynamics. One important variable to be determined during aircraft measurements is the atmospheric temperature, ideally not only at flight level, as provided in high resolution by the standard aircraft instrumentation. For this measurement it is desirable to use a remote sensing technique,

which provides good horizontal and vertical resolution. One such instrument is the passive microwave temperature profiler (MTP; Denning et al. 1989), which provides temperature profile information both, above and below flight level. Due to its measurement principle, it is largely independent from the prevailing atmospheric conditions, such as clouds or sunlight, which hardly interfere with the measurement – an advantage of the MTP in comparison with other remote-sensing techniques, such as LIDAR instruments. Such an instrument has been purchased and deployed by the German Aerospace

Center (DLR).

Since its invention in the late 1970ies, the MTP has been deployed in a number of aircraft campaigns (Mahoney and Denning, 2009), and continues to be developed to meet today's standards of technical requirements, and data recording. The MTP records thermal radiation emitted by oxygen molecules in the atmosphere. Its level-1 data output is a brightness

temperature, which has to be converted to absolute temperature profiles by using a retrieval algorithm, that utilises forward radiative transfer calculations. For correct interpretation of the structures found in the retrieved temperature fields, it is necessary to have precise knowledge of the instrument characteristics, such as the instrument response function, and the precision and accuracy of the brightness temperature measurements that are the input to the retrieval algorithm.

MTP data have been used for a variety of purposes, both directly interpreting MTP data or using it to support studies of data from other instruments or model output, as the following brief overview will show. MTP data has been used to interpret in situ measurements of trace gases (e.g. Marcy et al., 2007; Thornton et al., 2007; Spinei et al., 2015), aerosols, (e.g. Gamblin et al., 2006; Popp et al., 2006; Schwarz et al., 2008), and assist the study of cloud physics (e.g. Corti et al., 2008; Jensen et al., 2010; Schumann et al., 2017; Urbanek et al., 2017), and dynamics in the atmosphere (e.g. Tuck et al., 1997, 2003;

Dörnbrack et al., 2002; Sitnikova et al., 2009). Studies focussing exclusively on MTP data include the derivation of the boundary layer height from MTP potential temperature isentropes (Nielsen-Gammon et al., 2008). After the Airborne Antarctic Ozone Experiment (AAOE), Hartmann et al. (1989) calculated potential vorticity on potential temperature isentropes from MTP measurements combined with pressure and wind measurements provided by the meteorological measurement system of the aircraft to gain insight in mixing processes within the polar vortex. Davis et al. (2014)

investigated the cold point temperature and mesoscale temperature fluctuations, derived as the difference to the mission average temperature, in the upper troposphere and lower stratosphere (UTLS), in connection to tropical weather disturbances.





Furthermore, MTP measurements have been utilised to investigate gravity waves (GWs) in the atmosphere. Studies focussed on general overviews (Gary, 2006, 2008), the formation of polar stratospheric clouds (PSCs; Murphy and Gary, 1995 and Tabazadeh et al., 1996), or the characterisation of gravity waves encountered during flight (Gary, 1989, Chan et al., 1993, Dean-Day et al., 1998, and Wang et al., 2006 ). Based on these mesoscale temperature fluctuation analyses, a number of

modelling studies have been performed and compared to cases in which MTP measurements are available. These studies aimed at improving the understanding and numerical description of GWs. Such studies have been undertaken by Bacmeister et al. (1990, 1996, 1999), Pfister et al. (1993), Cho et al. (1999), Leutbecher and Volkert (2000), Dörnbrack et al. (2002), and Eckermann et al. (2006).

Especially for these studies focusing on mesoscale temperature fluctuations, precise knowledge of the instrument

characteristics, such as intrinsic noise, and precision of the measurements is necessary, when identifying potential gravity wave signals within the time series of MTP data. Knowing the true range of sensitivity is also necessary to understand the shape and characteristic structures within the retrieved temperature profiles.

Despite the consistent use of data from various MTP instruments in many studies over the past decades, a thorough

instrument characterisation and derivation of measurement accuracy has not been published. For the first time, this study presents all those relevant instrument characteristics for an MTP instrument. This is the foundation for correct analysis and interpretation of data recorded by the DLR-MTP. This includes measurements of the instrument response function, the antenna diagram, and other inherent characteristics, such as measurement noise (Section 3), discussing calibration strategies to determine the best practice (Section 4), including a discussion of the influence of flight level changes on the instrument

state and measurement performance. Finally, radiative transfer calculations are used to determine whether the current measurement settings are already ideal, discussing possible improvements ( Section 5). Furthermore, a brief description of the instrument and its measuring principle will be given in Section 2. The findings are summarised in Section 6.

## 2 Instrument description

The MTP was developed in the late 1970s by Bruce Gary and Richard Denning at the Jet Propulsion Laboratory (NASA-

JPL) for research on clear air turbulence (CAT; Gary, 1989). Since its first deployment in the Stratospheric-Tropospheric Exchange Project (STEP) in Australia, 1987, the MTP has widely been regarded as an instrument providing valuable background information on the state of the atmosphere.

The latest development is the MTP as wing-canister instrument (see Figure 1, left panel), which can be mounted underneath the wing of a research aircraft (e.g. Haggerty et al., 2014). Two such MTP instruments have been built, and one of them has

been acquired by the German Aerospace Center (DLR).



The concept of measurements of the MTP as a passive total-power radiometer (Denning et al., 1989) is straightforward. As all radiometers measuring atmospheric temperature, the MTP uses absorption lines of the 60 GHz oxygen complex ('V-band'), which are caused by magnetic dipole transitions (Liebe et al., 1992). Passive radiometers pick up the energy transported by the photons emitted in these transitions. In this part of the spectrum, the Rayleigh-Jeans relation can be used

to describe the source function of the radiance picked up by the MTP:

$$B(\nu, T) = \frac{2h\nu^3}{c^2} \cdot \frac{1}{\exp\left(\frac{h\nu}{k_B T}\right) - 1} \cong 2\frac{\nu^2}{c^2} \cdot k_B T \quad (Eq.\,2.1)$$

implying a linear relationship between the measured radiance $B$ and the temperature $T$ of the black body source at a certain frequency $\nu$, using the Planck-constant, $h$, velocity of light, $c$, and the Boltzmann-constant $k_B$. This temperature, $T$, is referred to as brightness temperature (BT).

## 2.1 Basic MTP instrument components

Table 1 lists all important parts of the MTP instrument used to record the MTP data during flight. A horn antenna is used as the receiver for the incoming atmospheric radiation. Through down-conversion with a defined frequency, the local oscillator frequency (LO), and low pass filtering only the part of the incoming radiation spectrum around the current LO is let through. The passing signal is converted to a voltage, which is proportional to the squared input intensity. This voltage is finally

translated to a digital count number, stored in the MTP data file, and later translated into a brightness temperature through calibration (see Section 4).

Using a rotating mirror in front of the instrument's antenna (number 3 in right panel of Figure 1), the direction from which the radiation is collected can be changed, making the derivation of altitude-resolved temperature profiles possible. In its standard deployment settings, as programmed in the JPL instrument software, ten viewing angles are being used during one

measurement cycle; five above the horizon, four underneath, and one pointing exactly towards the horizon. At each angle, measurements at three LOs, corresponding to the frequencies of three strong oxygen absorption lines, are made (see Table 1), before moving to the next elevation.

The temperatures of the important parts of the radiometer, such as the mixer, synthesizer, as well as the electronics are stabilised, to minimise the influence of the changing conditions during a research flight on the instrument state, and to

protect the electronic parts from malfunction due to condensation.

Furthermore, a calibration target is built into the instrument (number 2 in right panel of Figure 1), to which the mirror points after each cycle of atmospheric measurements. The target itself consists of carbon-ferrite on an aluminium plate, which is heated to a constant temperature (approximately 40 °C) using two conventional power resistors. The calibration target is surrounded by a 1-inch-thick Styrofoam insulation which is transparent for microwave radiation. The signals recorded while

pointing towards the heated target are combined with a noise diode (ND) signal and used for calibration (see Section 4) to



convert the measured signal to a BT, which is the temperature, an ideal black body would have, which emits the measured radiance, according to Eq. 2.1.

This BT is usually not equal to the outside air temperature, since the measured signal is influenced by multiple altitude layers of the atmosphere. To derive absolute temperature from the radiation measurement, forward radiative transfer calculations

have to be carried out and compared to the measured radiances. This happens in a retrieval algorithm that is applied in post-processing.

The instrument characteristics presented in the following sections of this work all correspond to the raw measurements or the brightness temperatures, which are input to such a retrieval algorithm. Retrieval methods and related uncertainties will not be discussed.

**2.2 Wing-canister instrument hardware characteristics**

The design of the DLR-MTP was first introduced in 2008 and allows mounting the MTP inside a canister underneath the wing of a research aircraft. Details of the instrument design can be found in Mahoney and Denning (2009). The wing-canister instrument is not the first design of the MTP. Differences to older instrument designs are mentioned in Lim et al (2013) and Haggerty et al. (2014). The most important upgrade is that the LO is now defined as a frequency near (or ideally

at) an oxygen absorption line centre so that the two flanks that are measured belong to the same line. The instrument is pointing forward, measuring the temperatures of air masses in front of the aircraft. The standard LOs and elevation angles used in a measurement cycle are summarised in Table 1. Two instruments were built using this new design. One has been deployed on the NCAR GV since 2008 (e.g. Lim et al., 2013; Davis et al., 2014; Haggerty et al., 2014), the other (hereafter referred to as DLR-MTP) was acquired by DLR, and has been flown on the German research aircraft HALO (Krautstrunk

and Giez 2012).

The filter band-with of the DLR-MTP is fixed to ±200 MHz around the LO, with a gap at the line center. However, the radiometer architecture using a mixer to down-shift the incoming signal allows measurements at various frequencies, depending on the chosen LO. The synthesizer used to generate the LO can be tuned between 12 GHz and 16 GHz. The

output signal is doubled twice, allowing for a frequency range of 48 GHz to 64 GHz for atmospheric measurements.

Two significant modifications to the original instrument were made by DLR: An embedded computer and an inertial measurement system including a Global Positioning System (GPS) antenna. In the original set-up a Visual Basic software package was provided by NASA-JPL to run the instrument during research flights. With the on-board computer and

integration of the inertial sensor, this software was translated to a LabView code, which was adjusted to use the additional data provided by the inertial sensor. With those modifications, the DLR-MTP can run autonomously, i.e. independent from a connection to a cabin computer, which is still provided, and can be used, e.g. to adjust settings during research flights using the HALO LAN network.



The DLR-MTP was first deployed during the Midlatitude Cirrus Experiment (ML CIRRUS) in 2014 (Voigt et al., 2017). The focus of this mission was to probe natural cirrus clouds as well as contrail cirrus throughout various stages of their life-cycles. The MTP was part of the wing-probe instrumentation and recorded data during all mission flights. In total, the DLR-MTP produced almost 63 hours of data during 13 mission flights, recording 17476 individual measurement cycles. Data from this campaign will be used to derive the DLR-MTP noise figure and in the investigation calibration methods in the following.

## 3 Characteristics of the wing-canister MTP flown on HALO

To retrieve absolute temperature profiles from the MTP measurements, radiative transfer calculations are carried out, to model the radiance a microwave radiometer would measure in a defined atmospheric state. To correctly do so, the instrument transmission function has to be known. This function is defined by the instrument's filter function, which defines which part of the recorded spectrum is used in data processing. Moreover, the antenna diagram shows how sensitive the receiver is to the different directions in the half-sphere it is pointing towards. Both those functions have been measured in a stable laboratory environment (Section 3.1).

Since this MTP instrument is mounted to the outside of the aircraft (c.f. Figure 1, left), the instrument experiences changes in surrounding pressure and temperature during measurement flights in which flight level changes can be quite common. They introduce changes in the instrument state, and can affect the characteristics of the measurement time series. Section 3.2 will summarise findings from laboratory tests to define the influence of temperature changes on the linear dependence of the recorded signal on the source temperature and overall measurement noise. Noise characterisation is particularly important when interpreting temperature fluctuation in a time series of MTP data. Only if the magnitude of the noise signal is known, significant atmospheric temperature fluctuations can be identified. Moreover, knowing possible periodicity in the noise signal is essential to distinguish between periodic temperature fluctuations, e.g. those caused by gravity waves, from instrument noise. For those characterisations the MTP was placed inside a temperature chamber (Figure 3) to simulate the changing outside air temperature during mission flights. The results are also compared to noise characteristics derived from ML CIRRUS mission data.

The effect of changing instrument state on the calibration of data will be discussed in Section 4.

### 3.1 Instrument function

The measurements of the instrument transmission functions, as well as of the antenna diagram, were made in a chamber completely covered in microwave absorbers. The MTP was installed on a rotatable platform. A tuneable signal source with a





horn antenna was placed in 5 m distance to the MTP instrument. The signal was then measured by the MTP, as well as by a power meter for reference. The source signal was chosen to have power so that the MTP signal was well over the inherent noise level. For the measurement of the filter function, the source frequency was tuned between LO – 300 MHz to LO + 300 MHz in steps of 1 MHz.

The measured signal is normalised and then corrected for frequency dependency, based on Friis Transmission Equation (e.g. Balanis (1997):

$$cnts_{corr} = \frac{cnts_{\mathrm{norm}}}{f^2/(min(f))^2} \quad (Eq.\,3.1)$$

Finally the signal power of the source, $P_{\mathrm{corr}}(f)$ , is taken into account in a final normalised signal representing the relative forward transmission, $cnts_{\mathrm{final}}$:

$$cnts_{\mathrm{final}} = \frac{cnts_{\mathrm{corr}}}{P_{\mathrm{corr}}(f)} \quad (Eq\,3.2)$$

The resulting instrument transmission functions for the three standard LOs are shown in Figure 2, left panel. It shows
symmetrical shapes for all LOs functions (i.e. radiances are recorded symmetrically from both flanks of the probed oxygen line), confirming a transmission of the signal between ±200 MHz around the LO (width of the plateau). The gap in the centre is created by the receiver architecture, using a double-side-band biased mixer.A certain 'waviness' is visible next to this gap. To exclude reflections from the chamber as a source, the measurements were repeated multiple times with slightly different positioning of the source antenna and the instrument. Since the results were similar in all measurements, the source of this
'waviness' is attributed to some internal source within the instrument.

The main result of measuring the antenna diagram is the field-of-view (FOV) of the instrument, defined by the full width half maximum (FWHM; red, dashed lines in Fig. 2 middle and right panel) of the antenna diagrams. It is actually mainly defined by the shape of the rotating mirror at the front of the instrument. The measurement was made using the same
laboratory setup as for the measurement of the transmission function. Both, the horizontal and the vertical plane were measured in steps of 1° rotation. The symmetric shape of the diagram implies that radiance is picked up equally strong from all directions. Note that the maxima of the side-lobes in the antenna diagrams have a maximum at -30 dB, meaning the signals from these spatial directions are 1000 times weaker than the signal picked up from the main viewing direction. The FOV is about 7.0°-7.5° in the horizontal and about 6.5°-7.0° in the vertical at all frequencies.

**3.2 Temperature dependence of MTP characteristics**

Changing surrounding temperatures of the MTP can influence the performance of the instrument: Amplifiers may change the relation between recorded signal and source temperature, despite the fundamental assumption in MTP calibration is that this relation is always linear. Moreover, the noise diode used for calibration may change its signal, and the overall instrument noise can be affected.



To investigate the dependence of instrument performance, a series of measurements inside a cold chamber was performed (see Figure 3). During this measurement series, the temperature of the cold chamber was successively lowered from 21 °C to -15 °C in steps of 5 °C. This temperature range resembles the temperatures the MTP experienced during its deployment in

the ML CIRRUS campaign in 2014, as shown in Figure 4, right panel. The pod air temperature sensor monitors the temperature inside the MTP's housing during the flight (c.f. Figure 1, right). In the cold chamber, the housing was not installed, to prevent over-heating of the instrument at higher temperatures. As a result, the readings of this sensor show the air temperature inside the cold chamber. The scanning unit temperature sensor keeps track of the temperature of the MTP instrument within close proximity to the crucial parts of the radiometer, such as the amplifiers or the mixer. The readings of

this sensor give an impression of the state of the instrument and its thermal stability. It can be seen that the response to lowering the cold chamber temperature is different between the two sensors. This is caused by the placement of the sensors, one being closer to some heated parts of the instrument, indicating that changes in the environment of the instrument are not equally influencing all parts of the instrument. Moreover, from the readings of the scanning unit temperature sensor (black line in left panel of Fig. 4) it can be seen that the MTP instrument takes some time to stabilise under the new temperature

conditions.

Along with the MTP instrument two microwave absorbers at ambient temperature (hereafter referred to as 'ambient targets'), and one microwave absorber submerged in liquid nitrogen (hereafter referred to as 'cold target') were placed in the chamber, in order to perform calibration measurements throughout the complete measurement series. The third type of calibration

target used in this measurement series is the built-in calibration target of the MTP instrument (see Section 2), hereafter referred to as 'hot target'.

Only those parts of the measurement series are used in which the scanning unit temperature is stable (the difference between two readings being smaller than an empirical threshold value of 0.04 K), to exclude effects from the instrument adjusting to

new environmental conditions. This adjustment can take up to 15 minutes after the initial temperature change. To ensure that results from the laboratory measurments are representative for instrument deployment on an aircraft, they will be compared to an analysis of mission data from the ML CIRRUS 2014 campaign.

### 3.2.1 Linearity of the sensor

Using the measurements of the two ambient targets installed within the chamber, it can be shown that for the DLR-MTP the

linear relation between the source temperature and the measurement output is given at all standard LOs (see Fig. 5). Since not only the temperature of the target changed during this test, but also the temperature of the sensor unit itself (see Fig. 4), it can also be established that the linear relationship between the measured signal and the source temperature is maintained





throughout changing conditions. The measurements corresponding to the two individual targets (different line colours in Fig. 5) are nearly identical, proving consistency of measurements.

The calibration parameters needed to calculate the brightness temperature ($T_B$) from the measured signal ('counts'; $c$) are

therefore the y-intercept (receiver noise temperature; $T_R$), and the slope of the line ($s_{cal}$), drawn through two points defined through measurements of calibration targets a known temperatures:

$$T_B = c \cdot s_{cal} - T_R \quad (Eq.\,3.3)$$

In Section 4 it will be shown that those parameters are depending on the instrument state, and can be related to housekeeping data representing the instrument state.

### 3.2.2 Noise characterisation

Using the same time-series of cold chamber measurements the instrument's noise figure was characterised, using the signal measured when pointing towards the hot target. When pointing towards a calibration target at a stable temperature, the mean measurement signal should not change over time, and the deviation from the mean represents the noise added by the instrument. An example of the measured signal while looking at the hot target during one measurement segment at constant cold-chamber temperature is shown as the grey line in Fig. 6.

Obviously, absolute stability can hardly be reached in a cold environment, while parts of the sensor unit are heated to approximately 40°C. Slight changes in system temperature over time have to be taken into account by applying a linear fit to the measured data of one segment (black line in Fig. 6) instead of simply subtracting the mean (blue line in Fig. 6). The resulting DLR-MTP noise figure, as shown in Figure 7 (top), can be characterised by a Gaussian distribution with a standard

deviation of approximately 6 cnts and the mean at 0 cnts.

The same method as for the cold chamber measurements is used for DLR-MTP data recorded during the ML CIRRUS campaign in 2014. Here, the criterion used to determine flight segments with nearly stable instrument states is a difference of the scanning unit temperature of less than 0.04 K between two cycles. Additionally, it was made sure that no altitude

changes were made ($\Delta z \leq 25$ m) or curves were flown during these segments. From all ML CIRRUS mission and test flights, 61 segments could be identified that satisfied the criteria and were at least 5 minutes long. The middle panel of Figure 7 shows the noise characteristics at LO 56.363 GHz. Plotted is also a Gaussian function with a mean at 0 cnts, and a standard deviation of 6 cnts, as implied by the cold chamber noise figure (green line). The results from the flight data evaluation are in excellent agreement with the values found in the laboratory environment, showing even smaller standard deviations of 5.2-

5.7 cnts, depending on LO. This is strong evidence that the DLR-MTP noise figure does not change between flights, and the laboratory characterisation can be used to determine long-term stability of the instrument in between campaigns.





For the spectral analysis of the noise figure the 61 ML CIRRUS mission flight segments are used again. Due to the varying lengths of the individual flight legs, the data is concatenated to a single time line for spectral analysis. The power spectrum of the noise signal at LO 56.363 GHz of the DLR-MTP, as shown in Fig. 7 (bottom), reveals that the measurement noise can best be described as a red noise, which is characterised by the auto-correlation α between a data point of the time series and

its precursors. According to Torrence and Compo (1998), the corresponding theoretical noise power spectrum for a range of wave numbers k, $P_k$, is given by:

$$P_k = \frac{1 - \alpha^2}{1 + \alpha^2 - 2\alpha \cos(2\pi k \,/\, N)} \quad (Eq.\,3.4)$$

For the three standard LOs, the lag-1 autocorrelation of MTP measurements during the ML CIRRUS campaign is $\alpha \cong 0.7$. Fit parameters characterising the noise figure at the three standard LOs are summarised in Table 2.

With the above findings, characterising the DLR-MTP noise figure as Gaussian-shaped, with mean at 0 counts, and a standard deviation of 6 cnts, as well as with the knowledge of the inherent periodic structure of the noise signal, it is now possible to determine whether periodic structures in a MTP temperature measurement time series are significant (high probability that they result from atmospheric temperature fluctuations), or noise-induced. Additionally, the standard deviation of the Gaussian distribution of noise values can be used to determine the variance of BTs derived from the raw

signals, once the calibration parameters are known.

## 4 Investigation of calibration methods for the DLR-MTP

In Section 3.2 it was shown that there is a linear response in the measured signal to changes in the source temperature, so that the measured signal can be related to a brightness temperature by using the linear relation of Equation 3.3.

While a line can be fitted through any two known points, which makes the calibration process very simple, the determination of the line parameters also bears the danger of inconsistencies under rapidly changing measurement conditions, which could lead to large errors in the calculated BTs. The cold chamber measurements described in the previous section are used to investigate the influence of the changing instrument state (due to changing surrounding temperature) on the calibration parameters and the ND signal. To determine a best practice for calibration of MTP raw data, various methods are being

tested.

For the DLR-MTP there are three possible calibration strategies that can be used to determine the line parameters:
1.   Hot-cold calibration, using a cold target (microwave absorber submerged in liquid nitrogen) at temperature $T_{cold}$ and an ambient target (microwave absorber at room temperature) at temperature $T_{amb}$ to derive the calibration
parameters, and using the equations:



$$T_B^{CCh}(c) = s_{cal}^{CCh}(c_{hot}) \cdot c - T_R^{CCh}(c_{hot}) \quad (Eq\ 4.1a)$$

$$s_{cal} = \frac{T_{amb} - T_{cold}}{c_{amb} - c_{cold}} \quad (Eq\ 4.1b)$$

$$T_R = T_{amb} - s_{cal} \cdot c_{amb} \quad (Eq\ 4.1c)$$

In which $c_{hot}$ denotes a system parameter that describes the instrument state (see following section), so that in-flight data can be related to laboratory measurements within a similar instrument state. This is the standard calibration method of radiometers in a stable environment. Using this method to calibrate the sensor, before making measurements in the atmosphere, provides the calibration parameters based on two temperatures which lie on the upper edge and below the expected measurement range. Thus, the validity of the calibration for the following measurements can be ensured, as long as the sensor itself is in the same surrounding conditions during the calibration as during the atmospheric measurements, and sufficient instrument stability is given.

Furthermore, this calibration method is necessary to characterise the noise diode signal used in the second calibration method, described below. However, since it makes use of external calibration targets, the calibration measurement can only be performed on the ground, where single calibration measurements at arbitrary room temperatures may not be representative of the instrument state during flight, as will be shown below. However, this method can be used to check the overall health of the instrument.

2. MTP built-in hot target (microwave absorber with a heated metal plate in the back) at temperature $T_{hot}$ combined with a noise diode offset signal $c_{ND}$

$$T_B^{ND}(c) = s_{cal} \cdot c - (T_{hot} - s_{cal} \cdot c_{hot}) \quad (Eq.\ 4.2a)$$

$$s_{cal} = \frac{T_{ND}}{c_{ND} - c_{hot}} \quad (Eq.\ 4.2b)$$

Using a noise diode to add a measurement signal, representing a known temperature difference to the temperature of the hot target, is the default way to calibrate MTP measurements. By using calibration measurements taken during flight, the calibration roughly follows the individual state of the instrument, whatever conditions the aircraft meets. The down-side of this method is that a faulty noise diode signal can jeopardise reliable calibration. Also, in this method two reference temperatures are used, which are above the expected measurement range. Hence small uncertainties in the determination of the calibration parameters may lead to large deviations in the calibrated data.

3. MTP built-in hot target combined with HALO static temperature (HALO TS), using the equation

$$T_B^{TS}(c) = s_{cal} \cdot c - (T_{hot} - s_{cal} \cdot c_{hot}) \quad (Eq.\ 4.3a)$$

$$s_{cal} = \frac{T_{hot} - TS}{c_{hot} - c_{0°}} \quad (Eq.\ 4.3b)$$

Here $c_{0°}$ represents the recorded signal at the horizontal viewing angle, which corresponds to the forward-looking measurement, probing the air masses directly in front of the aircraft. This method is an alternative to the previous calibration method, in the case that the noise diode signal cannot be used. It also follows the individual state of the



instrument during measurement flights, but since this method is using the HALO static temperature measurement, the MTP data are no longer independent from the aircraft measurements.

The following sections will show that each of the mentioned calibration strategies leads to comparable results, but differ in
their representation of the instrument state or the determination of uncertainty. A discussion at the end of the section will summarise the results, leading to an assessment of a best practice for calibration of DLR-MTP data after a campaign. This permits the best possible representation of the instrument state during flight, leading to the best possible measurement accuracy and estimation of uncertainty, before applying a retrieval algorithm to derive absolute temperatures.

**4.1 Hot-cold calibration in a cold chamber**

When performing cold-target measurements, the interference with a standing wave between the instrument's receiver hardware and the surface of the slowly evaporating liquid nitrogen was taken into account. As the DLR-MTP is a total-power radiometer (Denning et al., 1989), the output voltage of the detector is proportional to the square of the incoming intensity (Ulaby et al., 1981; Woodhouse, 2005). Thus, the times with least interference of the original signal and the
standing waves are defined by minima in the measured signal time series. To find those minima in the cold chamber measurement time series, several steps were taken:

(i) a running average ($N = 25$) is used to minimise the noise on the data; (ii) a spline-fit is used to find a smooth curve, representing the measurements; (iii) the fit is used to interpolate to a higher time-resolution; (iv) the minima of this interpolated curve are used to identify those individual measurement cycles closest to the minima in the time series on which
the calibration will be based. Due to noise, the calibration becomes more reliable, if a mean of more than one cycle close to a minimum in the time series is used, hence, the five measurements closest to the time of a minimum in the smooth curve are always included in the analysis.

The resulting calibration parameters are plotted over the corresponding scanning unit temperatures at the time the minimum
in the cold target measurements occurred. Fig. 8 clearly shows that the parameters do indeed change with the scanning unit temperature. That corroborates the assumption, that DLR-MTP flight data cannot simply be calibrated by using fixed calibration line parameters from laboratory measurements at single arbitrary room temperatures, since such measurements are only representative for specific instrument states. Still, it is possible to apply a linear fit to the data, providing a relationship between the MTP scanning unit temperature and the calibration parameters to be used at these temperatures. The
same is true when using the hot target measurement signal as a reference, which might better represent the current state of the instrument than the scanning unit temperature. The linear fit parameters are summarised in Table 3. Whether this is sufficient to represent the changing conditions during flight has to be tested using mission data (see Section 4.3).



## 4.2 Calibration using the MTP built-in target

When applying this (default) calibration method to MTP data, everything builds on the following two assumptions. The first is that the ND offset signal is the same each time the calibration measurements are performed. The second assumption is that the BT measured when pointing towards the heated target corresponds to the measurements of the temperature sensors at the
back of the target.

If one of those assumptions is incorrect, large calibration errors can occur due to the fact that the two points used to determine the calibration line parameters are both at quite high temperatures: The built-in calibration target is up to 100 K warmer than the outside air temperatures during flight, and $T_{\mathrm{ND}}$ is added to this temperature.

Both assumptions are tested in the following, using the calibration measurements performed in the cold-chamber.

### 4.2.1 Noise diode offset temperature

The ND offset signal has to be characterised using the hot-cold calibration method. To investigate the influence of changing conditions during a measurement flight, the cold chamber measurement series is used, during which the ND is repeatedly activated. Since the calibration parameters are already known from the hot-cold calibration, the temperature offset connected to the signal offset created by the ND can be calculated. Resulting DLR-MTP ND offset temperatures are shown in Fig. 9.
For better comparability, the means of the temperature and count values have been removed. Those correspond to the reference values in Table 4 (column 2 and 3). The ND offset temperature obviously depends on the count offset resulting from the induced noise on the input signal, which shows a clear dependency on the sensor unit temperature (colouring of the dots in Fig. 9).

Again, it is possible to apply a linear fit between the recorded ND offset signal, $\hat{c}_{\mathrm{ND}} = c_{\mathrm{ND}} - c_{\mathrm{hot}}$, and the associated ND offset temperature, derived from the hot-cold calibration method. This fit can be used to find the correct ND offset temperature required in the calibration of mission data. The linear fit values of this correction are shown in Table 4 (last column). In Fig. 9 the deviation of noise diode counts from the linear fit can be seen as being as large as 20 counts for any of the three LOs. This spread translates into the remaining uncertainty in the ND offset temperature.

### 4.2.2 Hot target temperature measurement

The housekeeping data of the MTP indicate large temperature differences between the air in front of the hot target and the heated back. This difference can reach up to 55K, so that temperature gradients within the absorber material could lead to a misinterpretation of the measured brightness temperature, since the calibration measurement is mostly influenced by the front of the absorber, of which the exact temperature is unknown. To investigate the hot target measurement characteristics
the calibration parameters, determined from the hot-cold calibration method, are used to calculate the hot target BT associated with the current measurement signal.



Indeed, Figure 10 shows the clear trend towards colder BTs with lower scanning unit temperatures, which correspond to a colder environment of the MTP instrument. This is contrary to the readings of two Pt100 temperature sensors, which shows the intended target temperature of the heaters placed at the metal back of the target, of just below 45°C during entire mission flights (orange line in Figure 10). The difference between the readings of the Pt100 sensors in the rear of the target and the

correct BTs measured during calibration can be as large as 3 K. Still, the linearity of the sensor again allows for a linear fit between the current scanning unit temperature and the average associated hot target BT. Thus, in-flight calibration can be performed, using a corrected hot target BT, according to the MTP instrument's housekeeping data. The parameters to correct the hot target BTs used in the calibration are shown in Table 5.

### 4.3 Calibration based on outside air temperature

During its deployment in the ML CIRRUS campaign in 2014, occasional failures of the ND, caused by a faulty soldered joint, were experienced. As the ND signal could not be used for calibration, HALO TS is used instead. This temperature is interpreted as the BT measured at the 0° elevation (horizontal measurement). In Section 5 it will be shown that the MTP measurements at all standard LOs are most sensitive to the air directly in front of the sensor (less than 2 km distance). Thus, the average HALO TS value of the 13 s - period it takes to record an entire MTP measurement cycle (with the 0°

measurement being in the middle of the cycle), is representative of the air masses probed by the 0° elevation measurements. Hence, the calibration parameters can be derived by using the calibration measurement while pointing at the hot target, combined with the horizontal measurement, using Eq. 4.3.

### 4.4 Comparison of calibration methods

There are eight different ways to perform the calibration calculations with and without applying the corrections discussed in

the previous sections, summarised in Table 6. All methods were compared to find the best practice of deriving BTs from MTP raw counts. All eight methods have been applied to the same set of mission data, using segments from all ML CIRRUS mission flights, during which the altitude of the aircraft did not change by more than 50 m between cycles, and no curves were flown (roll smaller than 5°). Note that this definition of usable legs is not based on any parameters connected to the DLR-MTP, leading to the inclusion of measurement cycles with possibly unstable measurement conditions, e.g. shortly after

altitude changes. The only exception is that only those segments are used, during which the ND did not show failures, to ensure comparability of all calibration methods. This way, 38 flight segments of at least 10 minutes length could be identified. The BTs are calculated based on each individual measurement cycle, but using the calibration coefficients ($s_{cal}$ and $T_R$) calculated from the average of the relevant data from the seven previous cycles, the seven following cycles, and the cycle itself (N = 15), to account for noise on the calibration measurement signals.

As an example, the resulting BTs of the 56.363 GHz measurements at 0° limb-viewing angle during one segment of ML CIRRUS flight MLC10 on April 11th, 2014, are shown in Fig. 11. For plotting purposes, the difference between the BTs derived with each individual calibration method to the HALO TS is shown. The BTs resulting from all calibration methods



show the same time-dependent variations, and mainly differ in their offset to HALO TS. This leads to the assumption that differences in the respective calibration coefficients affect the accuracy of the derived BTs more than the precision.

To further investigate the precision of the MTP measurements, a leg-mean value of the HALO TS and the BTs of the 0° elevation measurements is used to determine the offset, which is subtracted from the BTs at all elevation angles:

$$T_B^{corr}(v_{LO}, \alpha) = T_B(v_{LO}, \alpha) - \left( \overline{T_B(v_{LO}, 0°)} - \overline{TS} \right) \quad (Eq. 4.4)$$

with $T_B(v_{LO}, \alpha)$ and $T_B^{corr}(v_{LO}, \alpha)$ denoting the original and the corrected BTs under elevation angle α and at a specific LO, respectively. $\overline{T_B(v_{LO}, 0°)}$ denotes the leg-mean of the original BTs, measured under 0° elevation, and $\overline{TS}$ represents the leg-mean HALO TS. By using leg-mean values to determine the offset, the corrected BTs will still contain individual small-scale structures, which might differ from those in the HALO TS measurements. As a result, this correction will not influence further analysis of mesoscale structures. For individual calibration strategies, the subtracted offset can be as small as 0.8 K or as large as 7 K (see Fig. 11). The good agreement between all eight corrected BTs under the different viewing angles (see Fig. 12), indicates that removing the offset will not significantly change the shape of the temperature profile calculated in the retrieval. Moreover, the accuracy of temperature measurements now matches that of the HALO TS, which has an overall uncertainty of 0.5 K (Ungermann et al., 2015).

With the offset correction applied, plotting the RMS difference between the 0° BTs and HALO TS, shown in Fig. 13, gives a good impression of the capabilities of the different calibration strategies.

Naturally, the methods that make use of HALO TS show the smallest deviation from HALO TS readings. However, it is the intention to maintain an independence of the MTP measurements from other measurement systems, increasing the value MTP data adds to the package of instruments flown on HALO. Of those calibration methods that do not use the HALO TS, to the most reliable results are obtained when applying the method 'CCH', which uses the calibration values from the hot-cold calibration in the cold chamber measurements, related to the current hot target measurement signal. Whenever reliable ND measurements are available, this calibration method provides equally reliable results. Applying the corrections to $T_{ND}$ or $T_{hot}$ does not significantly change the result, but slightly smaller deviations from HALO TS are seen for the BTs derived using only the $T_{hot}$ correction (method 'TND1b') or both corrections (method 'TND2'). Considering the ND failures during the ML CIRRUS campaign, the favoured calibration strategy is method 'CCH', also applying the offset-correction between the leg-mean 0° BT and the leg-mean HALO TS. The deviation between the resulting 0° elevation BTs and HALO TS is ≤ 0.38 K at all three LOs for all ML CIRRUS flight legs with stable instrument conditions. This value is only exceeded when using the calibration method 'CCS', for all other methods it can be interpreted as the precision of MTP brightness temperature measurements, as will be shown below.

### 4.5 Discussion of uncertainties and measurement precision

To estimate the uncertainty of the calibrated BTs used as input to the retrieval algorithm, a number of contributions have to be taken into account. Some result from calibration parameter derivation, others are inherent to the instrument itself. Both will be discussed in the following.

### 4.5.1 Uncertainty resulting from calibration parameters

Uncertainties arise from the use of the different reference temperatures used in the calibration process and are summarised in Table 7. It is clear that the individual uncertainties assigned to each of the contributing values are not all independent. For example, the uncertainty of the y-intercept ($T_R$) directly follows from the uncertainty of the slope of the line, but is also directly influenced by changing instrument states. Hence, a quadratic sum of the individual errors is not suitable and will lead to a large over-estimation of the total BT error. Hence, a sensitivity analysis to estimate the overall uncertainty is performed: reference values (see last column in Table 7) for all parameters with uncertainties are used in a reference calculation. With these values, BTs are calculated for a range of counts between 17500 and 19725, which corresponds to the measurement signal range for atmospheric temperatures, as seen during the ML CIRRUS campaign.

Two control calculations are made, adding the corresponding uncertainties (see second-to-last columns in Table 7) in a way that the slope of the calibration line becomes as steep as possible ($s_{cal}^{max}$, red lines in Fig.14), or as flat as possible ($s_{cal}^{min}$, blue lines in Fig. 14), following:

$$s_{cal}^{max} = \frac{(T_1 + \Delta T_1) - (T_2 - \Delta T_2)}{(c_1 - c_2) - 2\Delta c} \quad (Eq\ 4.5a)$$

$$T_R^{max} = (T_2 - \Delta T_2) - s_{cal}^{max} \cdot (c_s + \Delta c) \quad (Eq\ 4.5b)$$

$$s_{cal}^{min} = \frac{(T_1 - \Delta T_1) - (T_2 + \Delta T_2)}{(c_1 - c_2) + 2\Delta c} \quad (Eq.\ 4.6a)$$

$$T_R^{min} = (T_2 + \Delta T_2) - s_{cal}^{min} \cdot (c_s - \Delta c) \quad (Eq.\ 4.6b)$$

assuming that $T_1$ (with associated measurement signal $c_1$) is the warmer temperature used in the calibration. Comparing the BTs of the reference calculation to those of the two control calculations reveals the maximum uncertainty in the derived BTs. Furthermore, in parallel to the offset correction introduced in the previous section, a BT correction for the control calculations is introduced: Here, the offset correction is calculated from the difference between the BTs of the control calculation ($T_B^{ctr}$), and that of the reference calculation ($T_B^{ref}$), at 18500 cnts:

$$T_B^{corr}(c) = T_B^{ctr}(c) - \left(T_B^{ctr}(18500) - T_B^{ref}(18500)\right) \quad (Eq.\ 4.7)$$

Results for all three calibration methods are shown in Figure 14. The shading around the two lines resulting from the two control calculations indicate the error range induced by 6 cnts uncertainty of the measurement signal that is to be calibrated (see Sect. 3.2.2 and Table 7). The vertical, grey-shaded region indicates the expected range of counts within one measurements cycle, if the horizontal measurement is at 18500 counts. Within this region, the resulting error, indicated by



the upper-most and lower-most edges of the blue- or red-shaded region, is comparable to, or smaller than the expected error

from the measurement noise itself ($\Delta T_B = \Delta c \cdot s_{cal}^{ref} - T_{Sys}^{ref}$), indicated by the horizontal black dashed lines. In that, the three

approaches to calibrate MTP measurements produce comparable uncertainties in the derived BTs. However, the calibration

method relating to the cold chamber measurements is most reliable in the case that the measured signals deviate largely from

the measurement signal at the horizontal elevation (i.e. if large vertical temperature gradients are present around the current

flight level). The overall uncertainty is clearly below the already established value of $\sim$ 0.38 K for all methods, mainly

caused by the measurement noise.

In the literature, a different approach to derive the measurement uncertainty, defined as the variance of measurement noise,

$\sigma_N$:

$$\sigma_N = \frac{T_{\text{sys}}}{\sqrt{\Delta f \cdot \tau}} = \frac{T_R + T_{\text{atmo}}}{\sqrt{\Delta f \cdot \tau}} \quad (Eq.\,4.8)$$

in which $\Delta f$ denotes the filter bandwidth, and $\tau$ represents the integration time (e.g. Ulaby et al., 1981; Woodhouse, 2005).

The DLR-MTP has an ideal filter width of $\Delta f$ = 400 MHz (see Fig. 2, left panel) and uses an integration time of 200 ms.

Assuming a receiver noise temperature of 493.79 K (see Table 7), and a mean atmospheric temperature of 250 K, this leads

to a theoretical value of $\sigma_{N,theo} = 0.0827\,K$, which is approximately four times smaller than the value established through

the calibration of mission data. However, values used to derive the theoretical variance do not take into account, that the

effective filter band width is smaller than the ideal value due to small deviations depending on frequency, and because of the

gap in the centre of the transmission function (see Fig. 2). Moreover, it does not consider gain fluctuations (Ulaby et al.,

1981). This is not representative of any real radiometric system that is applied outside controlled laboratory conditions,

especially the MTP. It also confirms that the uncertainty of derived BTs is dominated by gain fluctuations.

**4.5.2 Uncertainty from pointing error**

The position of the DLR-MTP instrument underneath the wing of the aircraft makes it sensitive to the altitude and speed of

the aircraft, which alters the pressure underneath the wing, leading to deviations between the assumed pointing of the

instrument and its true viewing direction. On the ground, this deviation offset can easily be determined. During flights, the

measurement of the inertial sensor, which is part of the DLR-MTP and constantly records the current pitch angle of the

instrument, is disturbed by the electromagnetic signal caused by the near-by mounted stepper motor, making the data not

reliable enough to allow for a real-time correction of the pointing of the MTP instrument. Thus, the real pointing has to be

determined after the flight. Analysing the few reliable data points available after the two campaign deployments revealed

that the relative deviation from the true horizontal plane was less than 1° -2° during entire mission flights. Compared to this,

the MTP's FOV of 7°-7.5° (see Figure 2 and Section 3.1) is clearly larger. Thus, it is safe to assume, that a deviation of the



elevation angle of 1° - 2° from the assumed angle does not have a considerable influence on the uncertainty of the retrieval input..

## 4.6 Summary

In the previous sub-sections the influence of changing outside air temperatures on the instrument state was investigated. A series of measurements in a cold-chamber was used to simulate in-flight conditions. It was shown that the linear relationship between the source temperature and the measured signal is maintained. Still, the measurements revealed clear changes in all calibration parameters, depending on the cold chamber temperature. This includes a change in the measured brightness temperature when pointing towards the built-in hot calibration target, as well as a change in receiver noise temperature caused by the electrical parts, and the calibration slope, caused by a change in amplification of the signal. Corrections to account for those changes have been found, and it could be shown, that with the application of those corrections, brightness temperatures could be derived from ML CIRRUS mission data, with an accuracy matching this of the HALO static temperature, and a precision better than 0.38 K. It has further been shown that the dominant source of measurement uncertainty is measurement noise.

## 5 Range of sensitivity

Since the MTP data has to be processed using a retrieval algorithm it is important to figure out the best possible input to this retrieval. The measurement settings needed for this ideal input are investigated in this section.

When analysing already published studies on data from older versions of the MTP, as well as from the sister-instrument to DLR's version, operated by NCAR, all using the output from the retrieval algorithm used by JPL, there are strong indications, that the range of sensitivity (i.e. the range within which the instrument is still able to pick up information on the state of the atmosphere) is smaller than implied by the retrieval output, which is given at an altitude range of ±8 km around the flight altitude of the aircraft.

The MTP flown on the ER-2 research aircraft at ~ 20 km altitude only has a measurement range of ~ ±2 - 3 km around flight altitude, while using LOs at similarly strong absorption lines to the current instrument (i.e. 57.3 GHz and 58.8 GHz; Gary, 1989). Likewise, the height range of the DC-8 instrument, with LOs at 55.51 GHz, 56.66 GHz, and 58.79 GHz has an 'applicable range' (within which the weighting function drops to 1/e) of roughly ±2.8 km (Gary, 2006). In their conclusion of analysis of data recorded with the NCAR-MTP Davis et al. (2014) mention that "it appears that more than about 3 km below the aircraft, the MTP may have difficulty identifying subtle mesoscale variations of temperature".

The range of sensitivity depends on two settings in the measurement strategy: The set of LOs at which the measurements are taken, and the set of elevation angles used. The latter influences the vertical resolution of the temperature profile that can be retrieved from the MTP measurements, while the choice of LO influences the altitude range that the measurement is





sensitive to. Within radiative transfer (RT) calculations this is determined by the transmission function, $\mathcal{T}(v) = \exp(-\tau(v))$, which describes the ration of outgoing and incoming radiation for a layer of the plane-parallel atmosphere. It is expressed through the optical depth $\tau(v)$ defined as the integral of the absorption coefficient $(\alpha)$, which depends on the frequency $(v)$, the length of the path $(s')$ through one layer, the pressure $(p)$ and temperature $(T)$ of the layer within the plane-parallel

atmosphere (e.g. Schreier et al., 2019):

$$\tau(v) = \int_0^s \alpha(v, s', p, T) ds' \quad (Eq. 5.1)$$

To investigate the range of sensitivity of the DLR-MTP, it is useful to calculate the signal contribution from each respective layer of the atmosphere, determined by the weighting function (WF) defined as:

$$W(v, s) = \frac{\partial \mathcal{T}(v, s)}{\partial s} = \alpha(v, s) \cdot \exp(-\tau(s)) = \alpha(v, s) \cdot \exp\left(-\int_0^s \alpha(s') ds'\right) \quad (Eq. 5.2)$$

The WFs for the three standard LOs used by the DLR-MTP under the nine non-horizontal viewing angles used in the

standard measurement strategy and assuming an aircraft altitude of 11 km are shown in Figure 15 (left panel). For RT calculations needed to derive the WFs, the Python scripts for Computational Atmospheric Spectroscopy (Py4CATS[1]; Schreier et al., 2019) are used. The WFs were computed from absorption coefficients using spectroscopic line parameters from high-resolution transmission molecular absorption database (HITRAN; Rothman et al., 1998), assuming a mid-latitude summer atmosphere (Anderson et al., 1986). To make them comparable the WFs are scaled so that the sum of weights for

each viewing angle and LO equals 1. The standard MTP WFs do not show any peaks away from the flight level, indicating that most information is gathered at the aircraft altitude. Nonetheless, from the difference between measurements under varying elevation angles and using different LOs, information on the vertical temperature profile can still be gathered. However, the weights at ±2 km distance to the aircraft are less than a tenth of those close to flight level, indicating that not much information is gathered at this distance or further away. At lower altitudes, with higher pressure leading to less

transmission beneath the aircraft, this distance is even less.

**5.1 Choice of LO frequencies**

Logically, the best idea to widen the range of sensitivity would be to use different LOs that are located at weaker absorption lines than the standard LOs, on the edge of the 60 GHz oxygen absorption complex or even between two lines, as was done

with the older MTP instruments. The choice of an LO at the center frequency of an absorption line has several advantages: (i) the symmetrical line shape makes the retrieval more exact, (ii) synthesiser errors (small derivations of the LO from the intended frequency) cannot lead to large errors (opposite to a placement in which a strong line is placed just outside the filter range), and (iii) pressure broadening has not as strong an effect as with a placement between two lines. Concerning the

---

[1] available at http://atmos.eoc.dlr.de/tools/Py4CAtS/



threshold of possible frequencies, water vapour absorption becomes important in RT calculations, whenever frequencies close to 50 GHz are used.

To test the influence of opacity of the atmosphere, radiative transfer calculations were made in which the temperatures of the
atmospheric layers between ground and 110 km altitude were all set to 250 K. The simulation is made, using TIRAMISU (Xu et al. 2016), a retrieval algorithm developed to process MTP brightness temperatures, which uses the radiative transfer model GARLIC (Schreier et al. 2014). Simulations are made for the whole spectrum of frequencies between 50 GHz and 60 GHz with 0.01 GHz resolution. This range includes the three standard LOs already in use, but also eight weaker absorption lines (Liebe et al. 1992). Furthermore, the simulations were carried out assuming six different flight altitudes between 2 km
and 15 km, which is the ceiling altitude of the HALO aircraft. In this setup, the expected brightness temperature for a measurement is 250 K, unless the optical thickness of the atmosphere is small enough that the cold cosmic background is influencing the measurement, leading to a smaller brightness temperature. The more transparent the atmosphere is at any frequency, the colder is the simulated brightness temperature, and the atmosphere close to the aircraft only contributes to a small part of the measured signal.

The resulting brightness temperatures are shown in Figure 16. The left panel shows those at limb-viewing angle 0° (horizontal viewing direction) and the right panel those at +80° (near-zenith). The solid, black line in the left panel of Figure 16 shows that for any LO below 54 GHz the atmosphere becomes partly transparent even at the horizontal viewing angle. Hence, those measurements cannot be calibrated (or offset-corrected) using HALO TS, indicating, that only LOs at
frequencies above 54 GHz should be considered. The results for the near-zenith measurements (right panel of Fig. 16) indicate that the atmosphere is partly transparent for all possible LO frequencies at nearly all flight altitudes. Whenever this transparency is too strong, the signal measured at weak absorption lines while looking downwards could be dominated by the surface temperature, which might not be well-known. As a result, for adding LOs to the MTP measurement strategy, only three possible LOs are considered: Those corresponding to the oxygen absorption lines at 54.671 GHz, 55.221 GHz and at
55.784 GHz. The weighting functions of those three possible LOs under the standard set of elevation angles are shown in Fig. 15 (right panel). Obviously, the new LOs at weaker oxygen absorption lines are sensitive to a much wider range of altitude layers, especially below the aircraft. However, above the aircraft the weighting functions look similar to those of the standard LOs. This is due to the partial transparency of the atmosphere at these frequencies, indicated by low BTs in Fig. 16, combined with the fact, that the viewing direction points through a medium that becomes optically thinner with increasing
distance to the sensor.



## 5.2 Choice of elevation angles

When discussing the choice of the set of elevation angles to be used in the MTP measurements, the signal path through the atmosphere has to be considered. By hardware-design limitations, the range of MTP viewing angles is limited to ±80°. To consider a new, feasible set of elevation angles, it makes sense to compare the path lengths of all possible elevation angles $\alpha$
with the shortest possible path length through a vertical layer of the atmosphere at maximum elevation (±80° relative to the horizon):

$$l_{rel80°} = \frac{\cos(10°)}{\cos(90° - \alpha)} \quad (Eq.\,5.3)$$

The relative path lengths to the ±80◦ angle are summarised in Table 8. Especially the three largest elevation angles used in the standard MTP measurement strategy (underlined values in Table 8) do not differ much in their path lengths. This can result in the WFs of different measurements being very similar (overlaying lines, e.g. below aircraft altitude in Fig. 17);
those measurements are (partly) redundant.

To derive a new set of elevation angles for MTP measurements with as much independent information as possible, a rule of thumb is used, that with each new angle the length of the signal path at 80° should be added, meaning that $l_{rel80°}$ is close to an integer. Corresponding rows are highlighted in grey in Table 8, including one angle with $l_{rel80°} \approx 1.5$. However, due to the fact, that the antenna beam of the MTP instrument has a field of view of 7° - 7.5°, the measurements at 11° and 14° would
overlap, and probably also not contain much different information from the measurement at 19°.

## 5.3 Determining a new measurement strategy

Using the findings of the above sections, redundancy in information within one measurement cycle of the DLR-MTP can be reduced. Moreover, at least below the aircraft, the range of sensitivity of the MTP measurements can be significantly enlarged by using at least one LO at the frequency of a weaker oxygen absorption line.
Since the MTP is mounted on a moving platform with approximate speed of 200 m/s, it is also necessary to consider the time it takes to record one complete measurement cycle. In favour of better horizontal resolution, the most appropriate set of elevation angles is ±14°, ±30°, ±41°, and ±80°, taking into account the field-of-view of the antenna. Thus, including the horizontal measurement, only nine elevation angles would be used, instead of the 10 standard angles, in which the down-looking set of angles is smaller than the up-looking set; leaving out the -55° limb-angle. Since the up-looking WFs of all
possible LOs are very similar, the opposite would be more feasible: using more down-looking angles to enhance the resolution of measurements below the aircraft, but reduce the number of up-looking angles, e.g. by leaving out the +41° measurement.

Based on all previous considerations, four new measurement strategies are proposed and summarised in Table 9. The new
strategies are compromises between vertical resolution and range of sensitivity, keeping the total number of measurements





per cycle close to the original, so the total time of recording a complete measurement cycle does not change significantly, keeping the horizontal resolution of measurements. All proposed strategies use eight viewing angles and four LOs to enhance the vertical resolution and altitude range at the same time. The weighting functions of the measured signals for each of those new strategies are shown in Fig. 17. Depicted are the cumulative weights to indicate the percentage of the

measurement signal that is acquired with increasing distance to the aircraft. This depiction helps to understand how much a certain layer of the atmosphere contributes to the total signal at a single viewing angle and frequency. If two lines in the figure over-lap, the corresponding measurements (i.e. measurements at two certain frequency- and viewing angle combinations) are redundant. To compare the relative contributions of different frequencies to the total incoming signal, please refer to Fig. 16, where absolute values are shown.

Strategy '8E4LOa' shows the result of simply adding a LO to the standard set (Fig. 17a). In the other three proposed strategies, only two LOs of the original set are kept, and two LOs at weaker absorption lines are added. Since it is desirable to have the least redundancy in the measurement, overlaying weighting functions, as seen, for example, in Fig. 17 d) are to be avoided. Also, if the aircraft is flying at lower altitudes, the LO at the weakest absorption line might be influenced by the surface temperature. Hence, depending on the planned flight pattern, strategies '8E4LOb1' (Fig. 17c)) or '8E4LOc' (Fig.

17b) should be favoured. Both strategies increase the MTP measurement sensitivity to a wider altitude range, especially below flight level, while allowing for a well-resolved temperature retrieval and keeping the horizontal resolution of MTP data.

## 6 Summary

This work shows a thorough investigation of the MTP instrument operated by DLR and flown on the HALO aircraft. It is the

first time a thorough characterisation of a MTP instrument is published. The results are a necessary basis for the analysis and interpretation of temperature profile data retrieved from MTP measurements, which are used to assess the state of the atmosphere around flight level of the research aircraft, and to investigate mesoscale temperature fluctuations in the atmosphere. The knowledge of the instrument characteristics, its calibration and related measurement uncertainties, as well as the region on which the instrument can collect information is fundamental to the correct interpretation of the measured

signals and conclusions about the atmospheric conditions.

Using the standard measurement settings, the instrument response function was determined along with the antenna diagram (Section 3). Both measurements are crucial parts of the retrieval algorithm used to derive absolute temperature profiles from MTP raw data. The results show symmetric shapes of all transmission functions, which is the desired and expected result.

While small side-lobes are detected in the antenna diagram, the main lobe has a symmetrical Gaussian shape, with a full-with-half-maximum of 7.0° - 7.5°, which represents the field of view of the instrument. A smaller field of view could only be





achieved by using a different shape of the rotating mirror as well as a larger horn antenna, which the compact design, needed for the wing-carrier instrument, does not allow for.

Laboratory measurements, as well as data recoded during a field campaign, were used to characterise the MTP's noise figure (Section 3). After removing a drift on the data, caused by the instrument adapting to changing surrounding temperatures, the
remaining noise signal can be described by a Gaussian distribution with mean of zero counts, and a standard deviation of 6 counts. It was shown, that the measurement noise can be characterised as a red noise, with lag-1 auto-correlation of 0.7, which indicates that a time-series of MTP data may show wave-like structures caused by internal noise. The presented characterisation of the DLR-MTP noise figure is sufficient to identify significant atmospheric signals in MTP measurement time series.

The laboratory measurements were also used to proof the linear relationship between the instrument's measurements and the source temperature. Based on this linear relationship, the calibration of MTP raw data to derive brightness temperatures is possible, and was further analysed in a series of measurements in a cold chamber (Section 4), in which the changing temperatures surrounding the MTP during a mission flight were simulated. An influence on the instrument state was indeed
found, so that calibration parameters change, depending on the surrounding conditions. However, linear relationships between the calibration parameters and the instrument state (expressed through the readings of a housekeeping temperature sensor, or the measured signal while pointing towards the heated calibration target), could be established, and can be used in the conversion of raw measurement signal to brightness temperatures. Furthermore, a similar correction was established to account for the temperature gradient present in the heated calibration target.

Three different possibilities to perform the calibration are presented and compared. One of them refers to the cold-chamber measurements, and proofs to be extremely reliable when tested with campaign data from the ML CIRRUS campaign. The other two methods utilise the built-in heated calibration target, and provide a more direct relation to the in-flight instrument state. The noise diode, used in one of those in-flight calibration methods, was characterised through hot-cold calibration. It was shown, that the noise signal depends on the instrument state (surrounding temperature), implying that a correction has to
be used when calibrating mission data. Again, a linear relationship was found that provides the relationship between the measured noise diode offset signal and the correct associated brightness temperature offset needed for the calibration.

The necessity of an offset-correction relative to HALO TS has been identified, both for the calibration relating to laboratory measurements, and that using the noise diode signal. The correction procedure was introduced as comparing the leg-mean of the calculated BTs at 0° elevation angle (horizontal measurement) to the leg-mean of HALO TS.

The offset-corrected brightness temperatures were compared to HALO TS using data from all ML CIRRUS mission flights. The RMS difference to the HALO TS is found to be between 0.25 K and 0.37 K. This range is valid for all calibration methods, as long as the ND functions properly. Otherwise the calibration methods using the ND offset signal cannot be applied and will lead to false results. It has been shown that the offset-corrected BTs of all calibration methods agree well



within this specified RMS range at all elevation angles used during the ML CIRRUS campaign in 2014. Hence all three calibration methods produce comparable results.

Considering the desire for MTP measurements mostly independent from other measurements (such as the HALO TS, which can then be used as reference), and technical problems with the ND, experienced during the ML CIRRUS campaign in 2014,

the favoured method of calibration is to use calibration parameters from the cold chamber measurement series, linked to the system state via the measurement signal while pointing towards the MTP built-in target. Using this method, BTs can be derived with a precision better than 0.38 K, and an accuracy matching that of the HALO TS measurements, which have an estimated overall error of 0.5 K (Ungermann et al., 2015).

When analysing the uncertainty of the calibrated brightness temperatures (Section 4.5.1), it was found that this method

performs best, whenever large vertical temperature gradients are present near flight level. Furthermore, the analysis of uncertainties of the calibration parameters shows that it is clearly dominated by the contribution from measurement noise. Other uncertainties, such as the pointing of the instrument, are negligible compared to this uncertainty.

In Section 5, improvements to the measurement strategy were proposed, after showing that the current standard settings do

not provide ideal input to the retrieval algorithm used in the processing of MTP data. While limitations to the range of sensitivity have already been indicated in earlier publications related to the MTP, a theoretical study involving radiative transfer calculations to infer the weighting functions for the MTP instrument has not been published so far.

It could be shown that using the standard settings the weighting functions of all measurements, under all elevation angles and local oscillator frequencies, indicate that the signal is mostly influenced by the first 1.5 - 2 km distance to the aircraft

altitude, both above and below flight level. The instrument hardly collects any usable information on the state of the atmosphere outside of the resulting ~ 3 km region around flight altitude (i.e. ±1.5 km around flight level).

A proposal to improve the measurement strategy for future missions of the MTP has been made, involving a reduction of the number of elevation angles used and including frequencies of weaker absorption lines of the 60 GHz oxygen absorption complex. The weighting functions connected to these new measurement strategies imply that the range of sensitivity above

the aircraft can be increased to at least 2 km, and up to approximately 4-5 km below the aircraft at an aircraft altitude of 11 km. At the same time the horizontal resolution of MTP measurements can be maintained. This is a significant improvement in the value of MTP data.

Overall, this work shows all necessary instrument parameters and characteristics needed to accurately analyse and interpret

the data produced by MTP measurements. It is the basis to understand measurement uncertainty, the (vertical) range in which derived atmospheric properties are valid, to identify significant atmospheric signals in times-series of MTP data, and a guideline for choosing the best-possible strategy to record and calibrate mission data. Using this information the best-possible data input for the retrieval algorithm, used to derive absolute temperature profiles, can be obtained. With that basis



the MTP can provide valuable information on the atmospheric state which can be utilised in many studies on atmospheric dynamics or in connection with in-situ measurements made on the same mission flights.

## Acknowledgements

This work was partly supported by the Bundesministerium für Bildung und Forschung (BMBF) under project 01LG1206C
(ROMIC/GW-LCYCLE). Fruitful discussions with Manfred Birk (DLR-MF), Martin Hagen (DLR-IPA), and Harald Czekala (RPG) have also contributed to the work presented in this study.

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





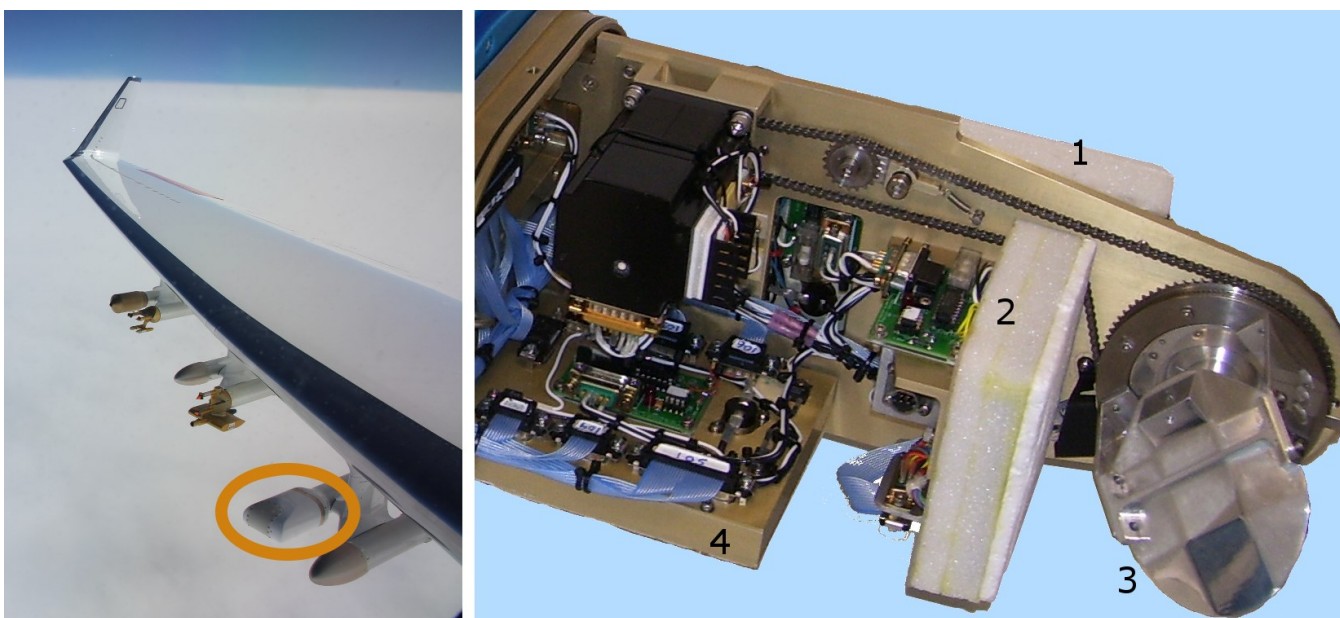

**Figure 1: DLR-MTP instrument. Left: Position of the MTP underneath the wing of the HALO aircraft. Right: MTP sensor unit in the lab. Marked with numbers are the radiometer unit (1), the hot calibration target (2), the rotating mirror (3) and the electronic unit (4), which contains various temperature sensors such as the scanning unit temperature or the pod air temperature sensor.**



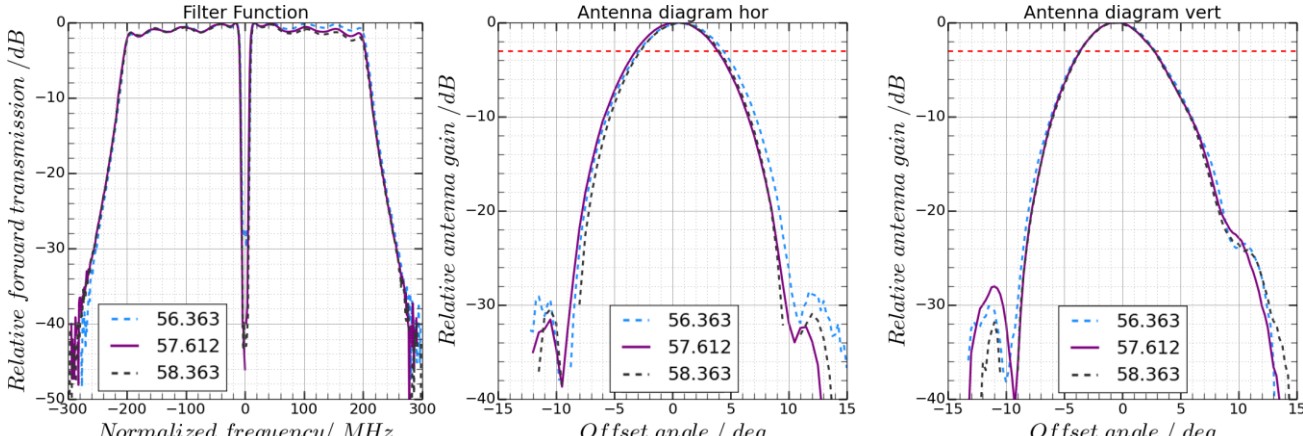

**Figure 2: DLR-MTP Filter functions (left panel) and antenna diagram of the horizontal (middle) and vertical (right) plane recorded at standard measurement frequencies. Red dashed lines indicate the half maximum value. All data are normalised so that the maximum value shown is 0.**





**Figure 3: DLR-MTP inside the cold-chamber. For the measurements, the box containing the liquid nitrogen and the ambient target was rotated to face towards the MTP sensor unit. A second microwave absorber was placed on the ceiling of the chamber to function as second ambient target.**



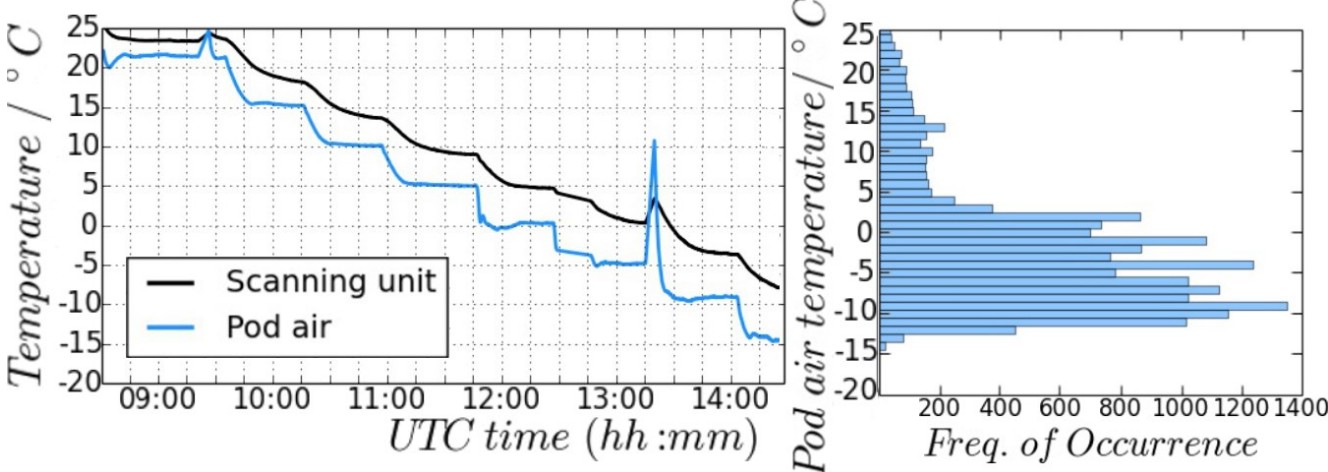

**Figure 4: Temperature sensor measurements during cold chamber measurements (left panel; black line: Scanning unit sensor, blue line: Pod air sensor). At 0 °C (~11:45 UTC) the cold chamber software had to be restarted, causing a longer stabilisation period, and at -5 °C (~13:15 UTC) the cold chamber was opened to re-fill the liquid nitrogen in the cold target causing the spikes in the temperature measurement. Right panel: Pod air temperature measurements during all ML CIRRUS campaign flights.**





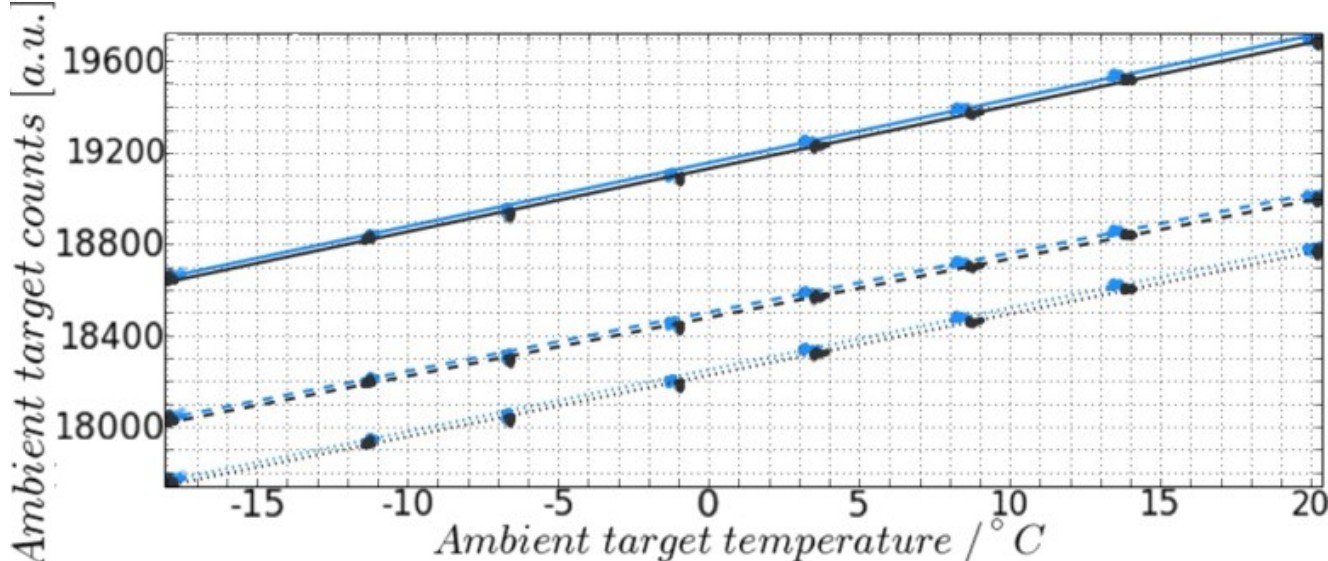

**Figure 5: Ambient target temperature vs. measured signal (counts) of the two ambient targets for all three standard LOs of the DLR-MTP (different line styles). Different line colours correspond to the measurement of the two individual ambient targets.**



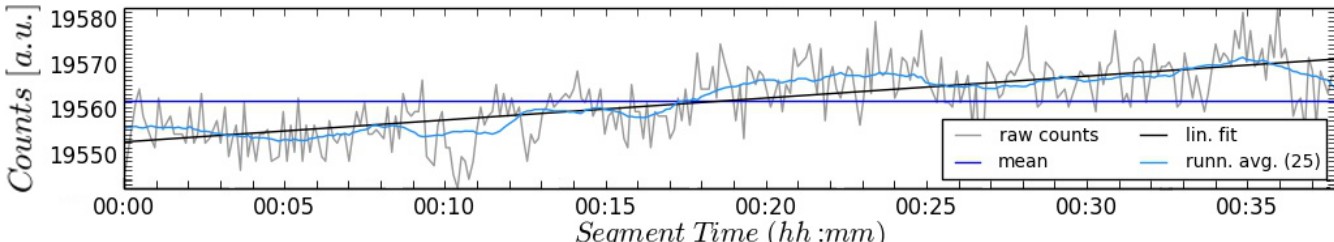

**Figure 6: Measured signal (grey line) at LO 56.363 GHz while looking at the ambient target inside the cold chamber as well as a running average (light, blue line), mean value (blue line) and linear fit (black line). The corresponding brightness temperature change in the linear fit during the segment is about 0.8 K.**

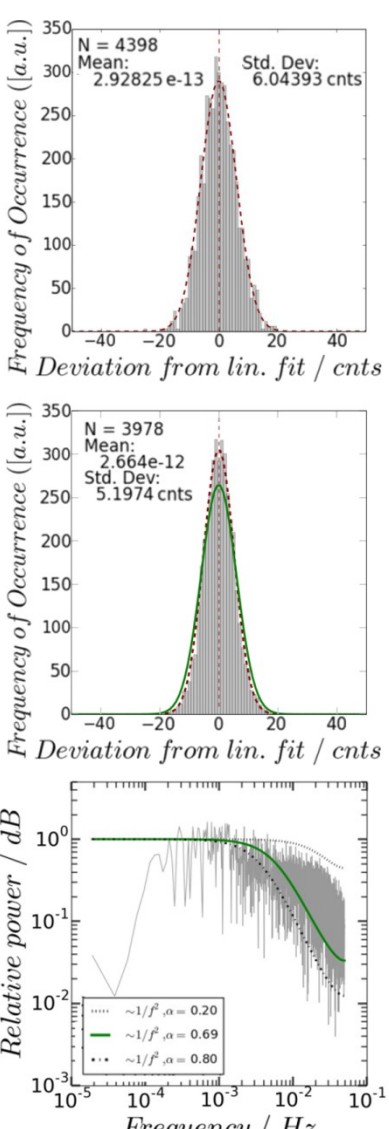

**Figure 7: DLR-MTP noise figure at LO 56.363 GHz. Figures of the other two standard LOs look similar. Fit parameters for all three LOs are summarised in Table 2. Red dashed line: Gauss-fit to data. Top: Laboratory measurements in cold chamber. Middle: derived from ML CIRRUS flight data. Green line: Ideal Gauss function with the mean at 0.0 cnts and 6 cnts standard deviation. Bottom: Noise spectrum calculated from ML CIRRUS flight data. Black, dashed lines: theoretical power spectra of $1/f^2$ noise with lag1-correlations of $\alpha = 0.2$ and $\alpha = 0.8$. Green, solid line: theoretical power spectrum of $1/f^2$ noise with lag1-correlation of input data.**



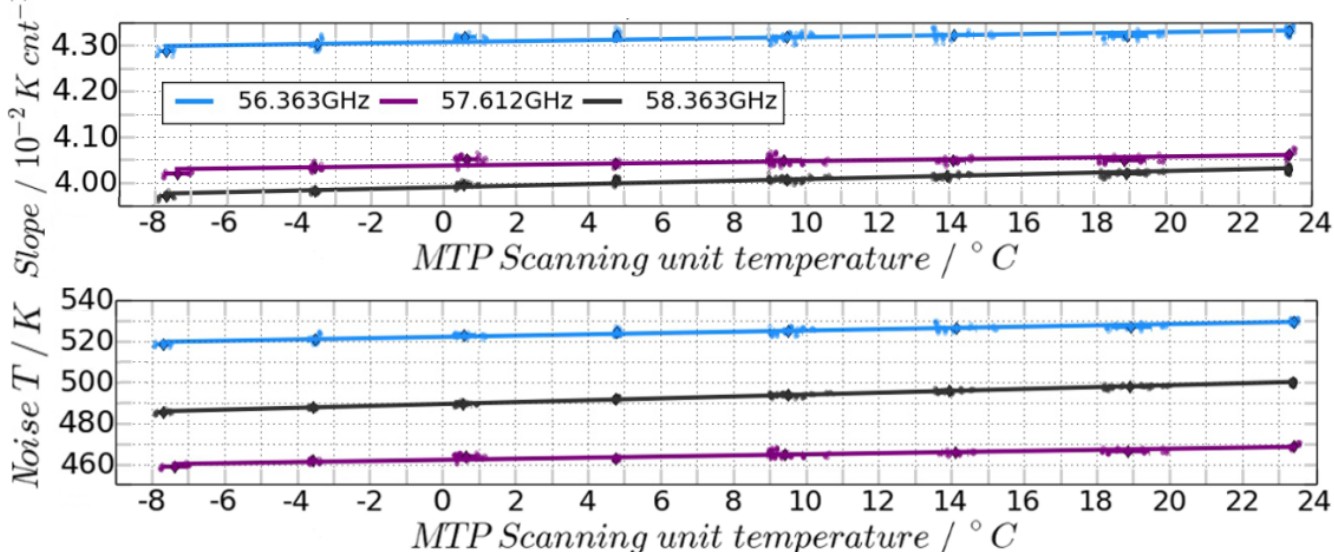

**Figure 8: Calibration parameters calculated from hot-cold calibration for standard LOs during cold chamber measurements. Top: slope of calibration line. Bottom: Receiver noise temperature ($T_R$).**





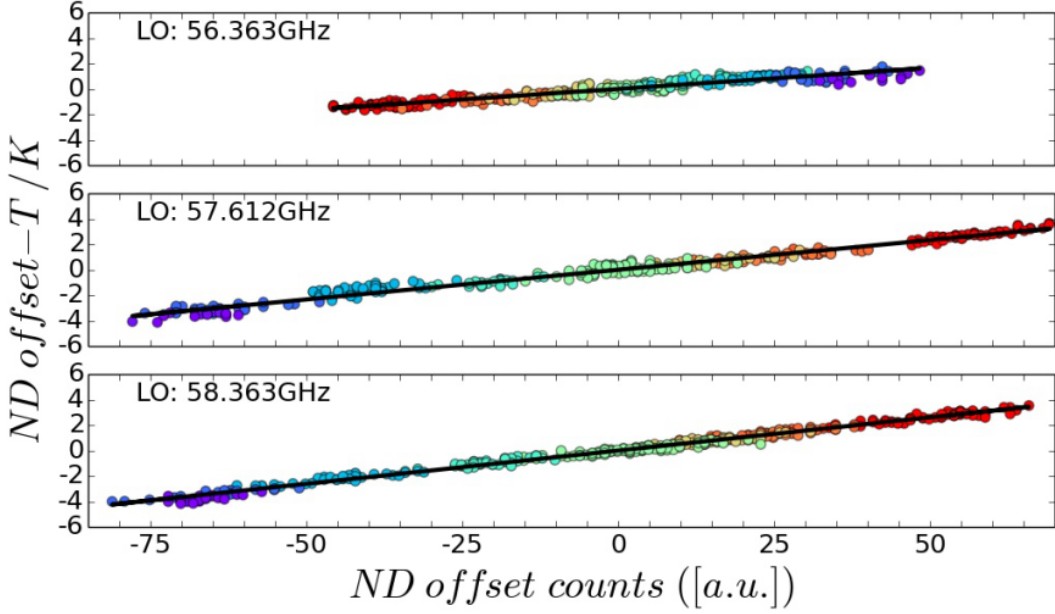

**Figure 9: Calculated ND offset temperature for occurring ND count offsets (mean removed from both values). Colour-coding: MTP scanning unit temperature (Blue: colder - red: warmer). Black line: Linear fit, linking ND offset temperature to offset counts.**





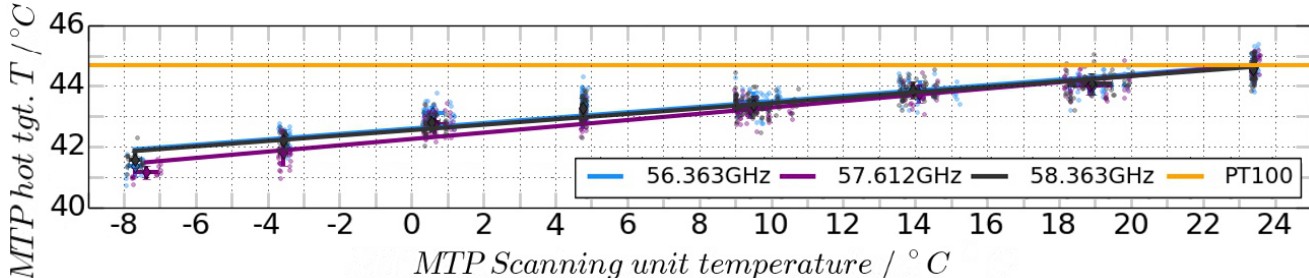

**Figure 10: Calculated hot target BTs at different scanning unit temperatures during cold chamber measurements. Small dots: Single measurements contributing to the average at one scanning unit temperature. Orange line: Pt100 sensor readings.**





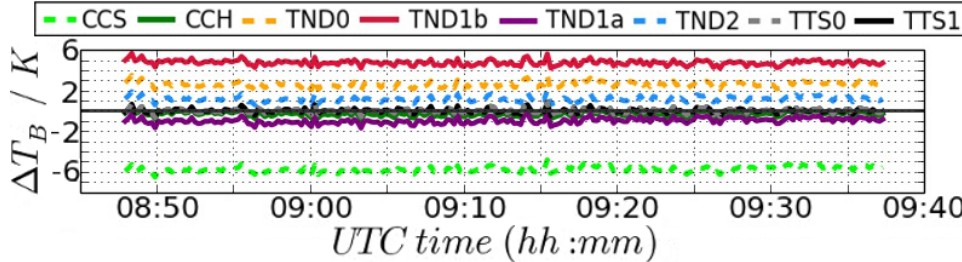

**Figure 11: Difference between BTs derived with the eight calibration methods defined in Table 6 and the BTs of 'TTS1' derived from the horizontal measurements at 56.363 GHz during one segment of an ML CIRRUS mission flight. .**





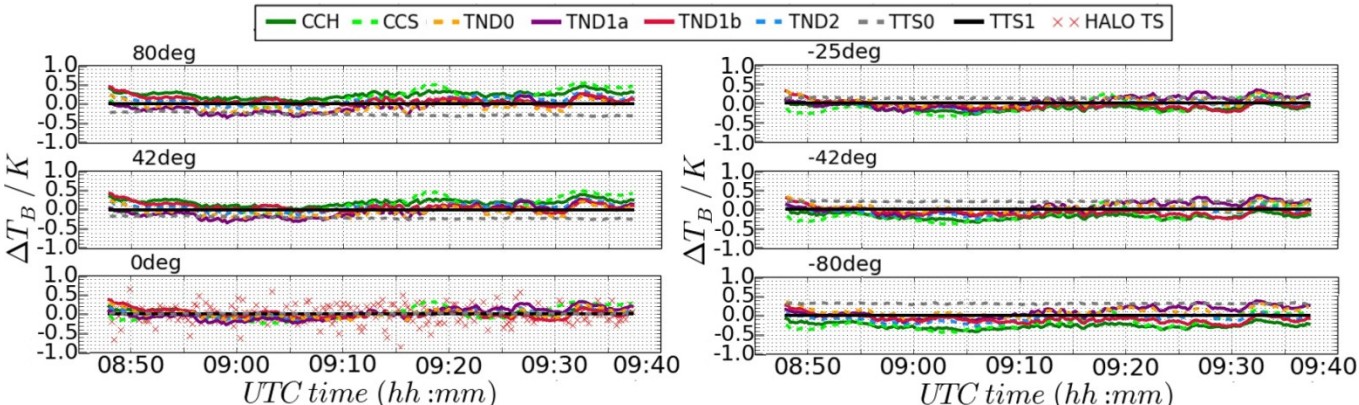

**Figure 12: Difference between offset-corrected BTs derived with the eight calibration methods defined in Table 6 and the offset-corrected BTs of 'TTS1' at six different elevation angles measured at 56.363 GHz. Red crosses in lower left panel: difference between method 'TTS1' and HALO TS.**





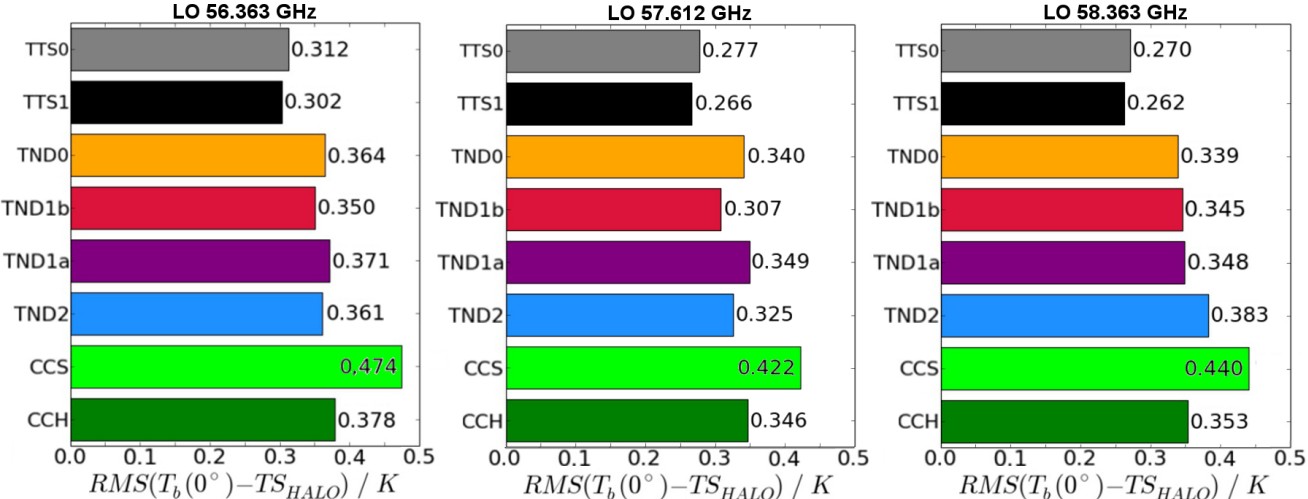

**Figure 13: RMS difference between HALO TS and BTs, derived from MTP measurement signal at limb-viewing angle 0° at the three standard LO during all ML CIRRUS flight segments with no altitude changes, curves, or ND failures, longer than 10 min. Refer to Table 6 for the denominations of the calibration methods.**





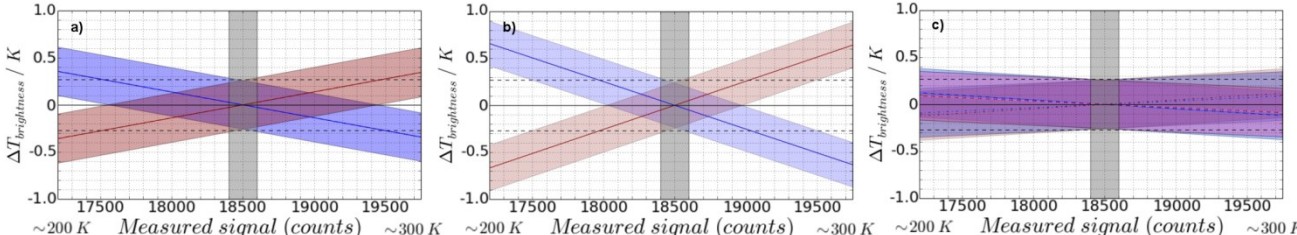

**Figure 14: Error estimation of calibration methods with applied HALO TS (assumed to be at 250 K) offset correction for calibration methods a) noise diode + hot target, b) hot target + HALO TS, and c) using cold-chamber calibration parameters. Vertical, grey shaded region: Expected range of measurement signals, if 0° measurement signal is at 18500 cnts (≈ 250 K). Black, dashed horizontal lines: Expected error induced by measurement noise of 6 cnts.**







**Figure 15: WFs of the MTP LOs (each individual line corresponding to a different viewing angle), calculated at aircraft altitude of 11 km. The $\bar{w}$ indicates averaging over all contributing frequencies within filter transmission range. Shown are the three standard LOs (left), and three possible LOs (right) to be used in a new measurement strategy of the DLR-MTP. Grey areas at the bottom: altitude range that would be below the surface.**





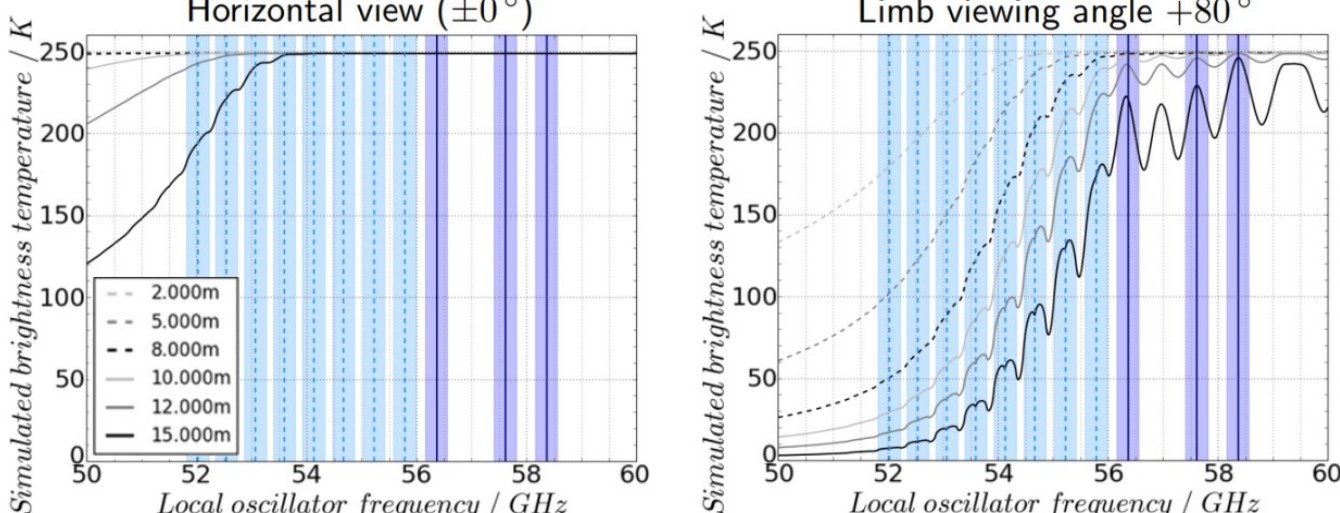

**Figure 16: Simulated BTs at LOs between 50 GHz and 60 GHz at different flight altitudes (different line styles) and at horizontal viewing angle (left panel), and at near-zenith angle (right panel). Solid, vertical lines: Standard LOs of the MTP; dashed vertical lines: strong lines that could be used as new MTP LO. Shading around vertical lines: MTP filter width.**



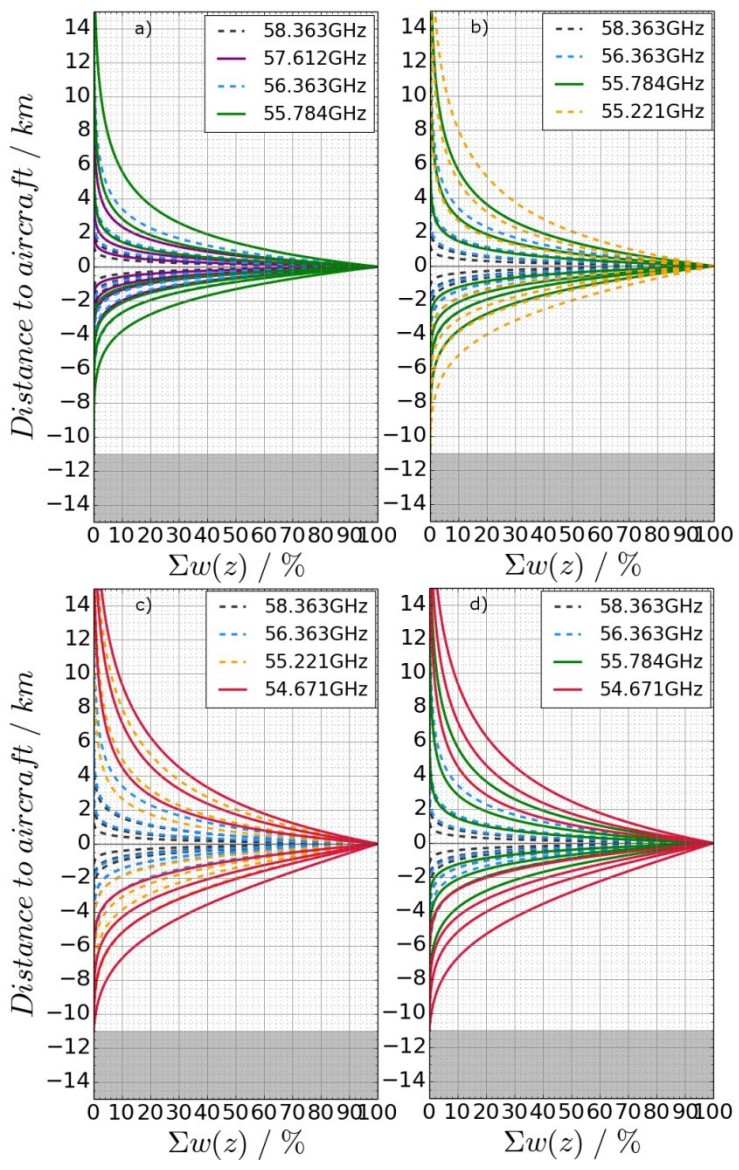

Figure 17: Cumulated weights of the MTP LOs (each individual line corresponding to a different viewing angle and frequency combination), calculated at aircraft altitude of 11 km. Shown are possible new measurement strategies, as mentioned in Table 9 a)'8E4LOa' b)'8E4LOc', c)'8E4LOb1', and d)'8E4LOb2'. Grey areas at the bottom: altitude range that would be below the surface.



| Component | Description | Standard settings |
|---|---|---|
| **Fairing** | Aluminium/ fiberglass to protect the hardware from the environment | |
| **Microwave window** | High-density polyethylene, allowing viewing of the atmosphere at ±80° range | |
| **Rotatable mirror** | Aluminium mirror designed to have a beam width of 7.5°; 360° rotatable. Steper motor used: Lin Engineering ,CE-5718L, step-size: 1.8° | +80°, +55°, +42°, +25°, +12° ±0°, -12°, -25°, -42°, -80° |
| **Horn antenna** | Conical, corrugated feed horn with an orthomode transducer (OMT) attached to the base | |
| **Radiometer parts** | Crossguide coupler for injection of a noise diode calibration signal Isolator to prevent local oscillator (LO) signal leakage. Double-side-band biased mixer Amplification and an intermediate frequency (IF) filter to select the pass band | Nominal filter band-with: ~ 200 MHz |
| **Frequency synthesiser** | Wide band YIG-tuned synthesizer with 1 Hz resolution can be tuned for an output frequency from 12.0 to 16.0 GHz The synthesiser output is doubled twice for a LO frequency range of 48 GHz – 64 GHz | 56.363 GHz, 57.612 GHz, 58.363 GHz |
| **Reference target** | 1-inch thick carbon-ferrite mounted on an aluminium plate Styrofoam and ROHACELL foam insulation (1/4 inch) Two conventional power resistors for temperature control; integrated in aluminium plate | Temperature control set to approx.. 41°C at the back of the target |
| **Data** | DC voltage proportional to the brightness temperature in front of the antenna, converted to digital counts | Recorded using LabView software on PC/104 |
| **Housekeeping** | Platinum Resistance Temperature Devices (RTDs) at various locations on MTP hardware | |
| **Controller PC** | Commercial ultralow-power single board computer in a PC/104 format with a passive heatsink Runs independent from cabin control | Connected to HALO network to enable user control if necessary or wanted |

**Table 1: MTP instrument: Components, and settings**



| | 56.363 GHz | 57.612 GHz | 58.363 GHz |
|---|---|---|---|
| **Gauss fit: lab measurements** **(N = 4398)** | Mean:  2.92825e-13 | Mean:  -1.5046e-12 | Mean:  -2.64535e-12 |
| | Std. Dev.:  6.04393 cnts | Std. Dev.:  6.0963 cnts | Std. Dev.:  6.2264 cnts |
| **Gauss fit: flight data** **(N = 3978)** | Mean:  2.664e-12 | Mean:  9.1452e-14 | Mean:  3.7587e-13 |
| | Std. Dev.:  5.1974 cnts | Std. Dev.:  5.1546 cnts | Std. Dev.:  5.72666 cnts |
| **Auto-correlation (spectral fit)** | $\alpha = 0.71$ | $\alpha = 0.70$ | $\alpha = 0.71$ |

**Table 2: MTP instrument noise characteristics at each of the three standard LO frequencies.**





| LO | Scanning unit temperature | | | | | Hot target counts $c_{hot}$ | | | | |
| | Ref. $T_{sc}$ | Ref. $s_{cal}$ | Lin. Fit slope | Ref. $T_R$ | Lin. Fit slope | Ref $c_{hot}$ | Ref $s_{cal}$ | Lin fit slope | Ref $T_R$ | Lin fit slope |
| GHz | °C | K cnts$^{-1}$ | $10^{-5}$ cnts$^{-1}$ | K cnts$^{-1}$ | $10^{-5}$ cnts$^{-1}$ | cnts | K cnts$^{-1}$ | $10^{-6}$K cnts$^{-2}$ | K cnts$^{-1}$ | $10^{-6}$K cnts$^{-2}$ |
|---|---|---|---|---|---|---|---|---|---|---|
| 56.363 | 7.518 | 0.043154 | 1.0937 | 524.492 | 0.3132 | 19486 | 0.043154 | 2.0141 | 524.492 | 0.0647 |
| 57.612 | 7.527 | 0.040446 | 0.9989 | 464.104 | 0.2716 | 19292 | 0.040446 | 1.8964 | 464.104 | 0.0559 |
| 58.363 | 7.474 | 0.040031 | 1.7775 | 492.777 | 0.4599 | 20213 | 0.040031 | 3.4361 | 492.777 | 0.0922 |

**Table 3: Linear fit values linking calibration slope values and receiver noise temperature $T_R$ (calibration *y*-intercept) to MTP scanning unit temperature and hot target counts.**





| LO [GHz] | ref. $\hat{c}_{ND}$ | Ref. $T_{ND}$ [K] | Lin. Fit slope [K cnts$^{-1}$] |
|---|---|---|---|
| 56.363 | 2799 | 120.90706 | 0.033089 |
| 57.612 | 3049 | 123.43799 | 0.046590 |
| 58.363 | 2932 | 117.53960 | 0.052118 |

**Table 4: Linear fit values linking noise diode offset temperature to MTP noise diode offset counts.**



| LO [GHz] | Ref. $T_{sc}$ [°C] | Ref. $T_{hot}$ [°C] | Lin. Fit slope [°C °C$^{-1}$] |
|---|---|---|---|
| 56.363 | 7.518 | 43.271843 | 0.89126 |
| 57.612 | 7.527 | 43.036542 | 0.103719 |
| 58.363 | 7.474 | 43.211868 | 0.088969 |

**Table 5: Linear fit values used to correct the MTP hot target brightness temperature.**





| | Laboratory parameters | | MTP hot target + noise diode | | | | MTP hot target + TS | |
|---|---|---|---|---|---|---|---|---|
| | 'CCS' | 'CCH' | 'TND0' | 'TND1a' | 'TND1b' | 'TND2' | 'TTS0' | 'TTS1' |
| Lab $s_{cal}$ | $T_{SC}$ | $c_{hot}$ | - | - | - | - | - | - |
| Lab $T_R$ | $T_{SC}$ | $c_{hot}$ | - | - | - | - | - | - |
| $T_{ND}$ | - | - | (u) | (c) | (u) | (c) | - | - |
| $T_{hot}$ | - | - | (u) | (u) | (c) | (c) | (u) | (c) |
| TS | - | - | - | - | - | - | (u) | (u) |

**Table 6: Calibration methods tested with MTP data. $T_{SC}$ indicates linking of the parameters to the scanning unit temperature, $c_{hot}$ indicates linking to hot target measurement signal. Usage of uncorrected data is denoted with a '(u)', applied corrections with a '(c)'.**



| Error source | Name | Estimation method | Uncertainty | Ref. value |
|---|---|---|---|---|
| Hot target brightness temperature | $\Delta T_{hot}$ | RMS to linear fit in cold-chamber measurements | 0.23 K | 315 K |
| HALO static temperature (TS) | $\Delta TS$ | RMS to 13s running average | 0.13 K | 250 K |
| ND offset temperature | $\Delta T_{ND}$ | RMS to linear fit in cold-chamber measurements | 0.25 K | 120.63 K |
| Cold-chamber slope | $\Delta s_{cal}^{CCh}$ | RMS to linear fit in cold-chamber measurements | $8.224 \times 10^{-5}$ K cnt$^{-1}$ | 0.04121 K cnt$^{-1}$ |
| Cold-chamber Y-intercept | $\Delta T_{R}^{CCh}$ | RMS to linear fit in cold-chamber measurements | 1.205 K | 493.79 K |
| Measurement noise | $\Delta c$ | Deviation from linear fit in stable flight segments | 6 cnts | 18500 cnts |

**Table 7: Individual uncertainties of values used in brightness temperature calculation.**





| $\alpha$ | $l_{rel80°}$ | $\alpha$ | $l_{rel80°}$ | $\alpha$ | $l_{rel80°}$ | $\alpha$ | $l_{rel80°}$ | $\alpha$ | $l_{rel80°}$ |
|---|---|---|---|---|---|---|---|---|---|
| 1° | 56.428 | 13° | 4.378 | 19° | 3.025 | 28° | 2.098 | 41° | 1.501 |
| […] | […] | 14° | 4.071 | 20° | 2.879 | 29° | 2.031 | 42° | 1.472 |
| 5° | 11.299 | 15° | 3.805 | […] | […] | 30° | 1.97 | […] | […] |
| […] | […] | 16° | 3.573 | 25° | 2.33 | 31° | 1.912 | 55° | 1.202 |
| 11° | 5.161 | 17° | 3.368 | 26° | 2.247 | […] | […] | […] | […] |
| 12° | 4.737 | 18° | 3.187 | 27° | 2.169 | 40° | 1.532 | 80° | 1.0 |

**Table 8: Signal path lengths relative to ±80°. Underlined: Elevation angles used in the standard measurement strategy. Grey cells: Possible candidates for a new strategy. Elevations in between (marked by '[…]') do not correlate with integer values in relative path lengths.**





| Name | Elevation angles | LOs [GHz] | $t_{cyc}$ |
|---|---|---|---|
| standard | +80°, +55°, +42°, +25°, +12°, ±0°, -12°, -25°, -42°, -80° | 56.363, 57.612, 58.363 | ~13 s |
| '8E4LOa' | +80°, +30°, +16°, ±0°, -16°, -30°, -41°, -80° | 55.784, 56.363, 57.612, 58.363 | ~14 s |
| '8E4LOb1' | +80°, +30°, +16°, ±0°, -16°, -30°, -41°, -80° | 54.671, 55.221, 56.363, 58.363 | ~14 s |
| '8E4LOb2' | +80°, +30°, +16°, ±0°, -16°, -30°, -41°, -80° | 54.671, 55.784, 56.363, 58.363 | ~14 s |
| '8E4LOc' | +80°, +30°, +16°, ±0°, -16°, -30°, -41°, -80° | 55.221,55.784, 56.363, 58.363 | ~14 s |

**Table 9: Proposed measurement strategies for future missions of the DLR-MTP.**