# Peer review of "Measurement Characteristics of an Airborne Microwave Temperature Profiler (MTP)"

_Atmospheric Measurement Techniques, 2019_

## Referee Comment (RC1) · Anonymous Referee #1 · 1 Mar 2020

The paper presents a characterisation of the airborne microwave temperature profiler (MTP) which is flying on the HALO research aircraft. It briefly investigates the spectral and antenna characteristics while the core of the paper is dedicated to identify the best calibration technique. In this way it addresses questions on instrument performance important for users of the airborne measurements and fits well to AMT. However, I see several weaknesses which need to be addressed before publication especially since the authors seem to have not much experiences with microwave radiometry. My main concern is that the different lab measurements/ analyses show little structure and are not clearly connected to the overall uncertainty assessment (see below for details) of the measured brightness temperatures.

I agree with the authors that the limitation to measurements space (brightness temperatures, TB) – leaving retrievals out – is sensible. Section 5 which is addressing an optimized scanning strategy for future flights seems unnecessary at this point as the rest of the paper deals with past measurements especially as it neglects several). In light of the already rather lengthy manuscript I suggest to leave this aspect out and instead relate the lab measurements to TB uncertainty using radiative transfer calculations. Furthermore, the authors should use the well established Allan variance technique for assessing the noise characteristics. My concerns are detailed below as well as many instances where the paper needs clarification.

General Comments:

1. The implications of the instrument characterisation for the subsequent interpretation of TB and temperature retrieval are not thoroughly assessed. Section 5 should have a clear outcome on the questions: (i) spectral characteristics: Which are the representative frequencies of the three channels? Which frequencies shall be assumed for the retrieval algorithm? Does the RT have to consider the full bandpass characteristics? (ii) Which is the effect of the antenna bandwidth? Is a pencil beam approach justified? (III) What noise characteristics have to be assumed in the retrieval, e.g. in the measurement covariance matrix?

2. Accurate calibration is the most important task in microwave radiometry. As all components are strongly temperature dependent besides temperature stabilisation a periodic calibration is needed. The calibration might only update the gain of the system (relative calibration) or make an absolute calibration in which all parameters of the raw measurement (count) to TB model are derived. In the simple linear case (as it is used in this manuscript) these are gain and receiver noise temperature Tr which can be derived by pointing the antenna successively to two reference targets. The authors seem to be not aware of this classical microwave formalism which is also apparent as they hardly cite any literature microwave radiometry (list in the back) and some flaws in the radiometer formula application. The major questions which would need to be addressed are: How good are the reference targets (blackbodies)? How frequently does a calibration need to be made? Why have the measurements in the cold chamber with view on a stable target not been used for such an analysis? The next step would then be the in flight calibration. Assuming that the laboratory calibration (strategy 1) would work is a bit naïve. However, there are good approaches later on using the horizontally pointing measurements but a motivation and explanation why this procedure was chosen needs to come first.

3. The information on the MTP measurement principle is not clearly provided in the beginning of the manuscript making it difficult for the reader to follow. Bits and pieces come together at different instances, e.g. scanning is explained on page 14 and especially the discussion on the use of different oxygen lines is confusing. For better understanding the authors should include a thorough description of the MTP measurement principle in the beginning and add an absorption spectrum (preferably even for different pressure levels as in Fig. 16) to illustrate the frequency channels (and their potential tuning range). This also serves to introduce the double sideband principle. Further, it could be explained why the LO is typically set at center frequency for mitigating problems due to frequency drifts, and how non-resonant emission (water vapor-continuum, hydrometeors) affects the measurement. This would also demonstrate that the LO frequency is not the frequency for which the measured TB is representative (passband averaged – see Fig. 16).

4. Section 5 address future measurement strategies in terms of frequency selection and elevation scanning. This is an important study but is not done as thoroughly as it is needed especially in light of vertical resolution of the retrieved temperature profiles for different types of atmosphere. It also does not take into account the findings of their laboratory measurements in respect to the spectral and spatial sensitivities. As the paper is already very lengthy it should be taken out.

5. At several instances it seems that the authors have gravity wave detection as application in their mind – this is ok but needs to be clearly stated (only abstract). Many

readers might not know which requirements in TB are needed for this purpose. Other users might be more interested in vertical resolution for stability assessment.

6. The readability of the paper needs to be improved - sometimes it is more a technical report than a paper. No clear goals are provided, the structure is not always clear, the text is written rather lengthy and many basic informations only appear rather late in a middle of a section where you would not expect it. Short paragraphs sometimes even only one sentence long occur and the text frequently repeats (unnecessarily) the figure captions, e.g. "Plotted is also a..". The paper could be shortened by reducing number of figures or using an appendix. I would recommend to concentrate only on the past measurements. The optimized scanning strategy In case but the future – which I think would be an own study if done carefully could go in an appendix.

Specific Comments: Why are brightness temperatures referred to as BT in the text (and Fig. 11) and TB in the equations. Historically the satellite community uses BT and the ground-based community TB. I don't think it matters which one is chosen but it should be consistent.

P1l8: "records radiances", no it records counts which are calibrated to brightness temperatures - it is ok to say TB here

P1l9: "state of the atmosphere can be derived" this indicates much more information than the temperature profile which was stated already – what else?

P1l22: "weaker oxygen lines" better write 'frequency channel'. The LO frequency of the channel does not necessarily need to be at a line center. Also and it seems to me that it is not clear to authors: the LO frequency is not the representative frequency of the channel – and the "representative frequency" can be extracted from their laboratory measurements. I anyway suggest to modify section 5 such that it can provide the necessary input for the retrieval algorithm

P1l22: "calibration parameters do clearly depend on the state of the instrument". This

is the key in microwave radiometry for astronomy, atmospheric, planetary science etc. ever since and for all instruments there is the question how frequently one has to calibrate, e.g. Dicke switching for short-term fluctuations. Unfortunately, even slight vibrations and temperature changes can cause transmission characteristics to change thus calibration parameters. So this sounds a bit naive – I recommend the authors to look more in basic microwave radiometry books, e.g. Janssen, 1994, Vowinkel, 2013, Woodhouse, 2017,

P1l26: Here it should be said that precision is determined for TB which closely relates to the atmospheric temperature when the instrument is pointed horizontally – otherwise it is confusing

P2l16. What is meant by structures?

P3l9-12 and P3l14: There is a very long list of applications of past studies using older versions of the MTP (is that really necessary?) and then it is claimed that instrument characteristics need to be known for correct interpretation. This is true and that's why this study is valid but it somehow implies that the work here also helps with data from old campaigns. This needs to be clarified.

Introduction: the whole introduction is dedicated to the MTP but there is no reference to other studies on the characterisation of other microwave airborne instruments is made, e.g. Blackwell et al, 2001 describing NAST with frequencies 50-57 GHz, Mc-Grawth and Hewison, 2001, Wang et al, 2007 etc. which might also check different instrumental parameters. The introduction clearly needs to mention the goals of the lab investigations.

P4l2: Not all radiometers for temperature profiling measure at the oxygen absorption complex around 60 GHz - also 118 GHz is used. In general, it is surprising that no reference is made to the fact that operational meteorological satellite instruments, e.g. AMSU-A, do temperature sounding since decades. These sounders exploit only the frequency information for profiling while the MTP aims at improving the resolution by angular information. It is necessary to explain the measurement principle here thoroughly, showing a spectrum (ideally for different altitudes) and the considered frequency channels. On a side note: The accuracy of the oxygen spectroscopy is still under debate which is, however, more important for retrievals, Caddedu et al, 2007; Cimini et al, 2018, Maschwitz et al 2013.

P4l2: Why don't you explain the heterodyne principle and talk about a double side band receiver. This is very important to clearly define the frequencies for the radiative transfer used for retrieval development.

P4l18: "making the retrieval of temperature profiles possible" Most instruments only use information on frequency dependence. Make clear that the MTP can achieve higher vertical resolution by adding the angular information.

P4l24: Thermal stabilisation is the most important part in a microwave radiometer the performance of all microwave components strongly depends on temperature. Therefore more details on that are needed.

P4l229: What about temperature stability, homogeneity, spill over of the target, cf. Mc Grawth and Hewison (2001).

P5, l14-15: the discussion on the oxygen spectrum and LO needs further explanation and should come before not in the section on wing-canister, same for the information on the frequency range (l25) below.

P5l22: how large is the gap, x MHz?

P6l 6 "investigation OF calibration"

Section 3. The frequency response of the bandpass is investigated but there is no discussion on the stability of the LO frequency – does this have any potential effect on measured TB?

P6l27: The authors mention the periodicity of the signal first. I understand that for

gravity wave detection this is important but in terms of radiometer performance the most important question is whether the instrument follows the radiometer formulae (Eq. 4.8), i.e. noise reduces with increasing integration time. For this purpose typically the Allan variance is used. This characterizes the noise and determines how long measurements can be integrated in time and how frequently a calibration needs to be performed.

Section 3.1: The name is irritating as it could mean much more. The measurements of the bandpass characteristics and the antenna diagram (section 3.1) are important and interesting but are presented rather briefly without any implications for the subsequent retrieval. Even the exact measured bandwidth and beamwidth are not given. For the analysis or implications RT calculation would play a major role. As for example shown in Crewell et al. (2012, their figure 10) the bandpass characteristics can cause the effectively measured TB being representative for a frequency deviating significantly from the specified channel frequency. In fact in the double side band approach this anyway takes place and needs to be handled in the RT underlying the retrieval process. Similarly, the antenna pattern smears out atmospheric features especially at low deviations from the horizontal in a vertically stratified atmosphere (Meunier et al., 2013). To appreciate this laboratory measurements and their impact on the measured TB further analysis is required which would fit well into section 5.

P7l12: "A certain 'waviness' is visible next to this" ripples are typical in any microwave component due to EM wave theory propagation – reducing the amplitude is key.

P7l23: how stable is the noise diode, how much does it depend on temperature (stabilization)?

P8l14: "takes some time to stabilize".. needs to be more quantitative – later it is mentioned but not here

Section 3.2: Information on the accuracy of the target temperatures is missing. P9l14 mentions the "hot" target – should be explained before.

P8l30: I find the term "at all LOs" confusing – also at other instances. Why not write for all frequency channels?

P9l7: Why do the authors not use the classical microwave notation using the gain (cf, Janzen, Mc Grawth and Hewison, ? The difference between receiver and system noise temperature needs to be made clear.

P9l17: Radiometers are never completely stable which is why periodic calibrations have to be made. In between this calibrations the TB could be corrected assuming a linear trend as shown in Fig. 6. The following paragraph describes this for the airborne measurements bit it is unclear for me that for these linear fits segments of 5 min without calibration are used?

P10l1 and following: The spectral analysis is interesting and similar to the Allan variance but is unclear to me why it is applied to atmospheric measurements and not to the cold chamber measurements where the real instrument performance could be tested. The concatenation eliminates real temporal signals. Does the analysis differ between in flight and laboratory measurements .

P10l20-27: "line parameters" is irritating as it could be interpreted in spectral lines: it is about the updating your calibration model, basically, gain and receiver noise temperature. It looks like the authors are not too familiar with typical microwave calibration techniques which is reflected by the lack of citation of microwave radiometer basics and studies. In operational receivers many strategies for that exist (Maschwitz et al., 2013) as typically gain needs to be adjusted more frequently than TR, relative/absolute calibration.

P11lEquation: Why so complicated $Tr^{CCh}(C\_hot)$ and not simply Tr – explain the meaning of the different indices.

P11l19: Give values to underline the statement

P12l4: The calibration strategies might serve different purposes. That the first strategy

leads to comparable results seems astonishing.

P12l12: The cause for the standing waves is the refractive index of the LN2 – here Küchler et al., 2016 should be cited for details. Here it sounds that just the evaporation is the reason

P12l25: Of course the calibration parameters change with changing environmental conditions if the temperature stabilization of the instrument is not perfect. The question to ask if this is repeatable. Would the same parameters be measured if the instrument had been moved and electrically disconnected in between?

P13l29: Why is the temperature unknown – more discussion is needed – see Mc Grawth and Hewison, 2001.

P16l8: why do you explain this only here and not at the beginning of the calibration section

P16l13: Nobody remembers counts better give the atmospheric temperatures and no-tate the counts with $c\_min$ and $c\_mac$ or later $c\_ref$ instead of 18500.

P16l24: "The vertical, grey shaded.." this is not paper style. The figure should be only a reference for the text.

P17l9: "In literature" then give a reference

P17l9 to 29: This paragraph shows that the authors have not much experience with microwave radiometry. It is weird to present the well established radiometer formula at the end and not in the beginning. The formula describes the internal noise of an ideal radiometer and typically one just writes a proportionality and not an equal sign as other losses occur (e.g. factor 2 for Dicke switching). Further, the authors put in 400 MHz as bandwidth but the double sideband receiver only has 200 MHz in the IF. The most important think to look at the radiometer formula is to check if the noise decreases with longer integration time which is basically what the Allan Variance technique does – it finds out at which point gain fluctuations dominate. This should be checked by the

laboratory measurements in the beginning and not in this section. Note, it is strange to only now to provide the integration time for atmospheric measurements.

P18l14: If the dominant uncertainty is the noise couldn't it be reduced by longer integration times?

P18l30: LO frequency

Table 1 does not include all instrument characteristics of interest, e.g. receiver noise temperatures, integration time, polarization. I am missing information on microwave window transmission

Fig. 8 could be combined with Fig. 10

Fig 11: Different calibrations need to be explained in figure caption. Caption does not say how the difference is calculated (what is the reference – the overall mean?). As I do not see significant temporal development mean and standard deviation could be just added as last lines in Table 6.

Blackwell, W. J., Barrett, J. W., Chen, F. W., Leslie, R. V., Rosenkranz, P.W., Schwartz, M. J., and Staelin, D. H.: NPOESS Aircraft Sounder Testbed-Microwave (NAST-M): instrument description and initial flight results, IEEE T. Geosci. Remote, 39, 2244–2253, 2001.

Cimini, D., Rosenkranz, P. W., Tretyakov, M. Y., Koshelev, M. A., & Romano, F. (2018). Uncertainty of atmospheric microwave absorption model: impact on ground-based radiometer simulations and retrievals. Atmospheric Chemistry and Physics, 18(20), 15231-15259.

Crewell, S., H. Czekala, U. Löhnert, C. Simmer, Th. Rose, R. Zimmermann, and R. Zimmermann, 2001: Microwave Radiometer for Cloud Carthography: A 22-channel ground-based microwave radiometer for atmospheric research. Radio Science, 36, 621-638, doi:10.1029/2000RS002396.

MP Cadeddu, VH Payne, SA Clough, K Cady-Pereira... Effect of the oxygen line-parameter modeling on temperature and humidity retrievals from ground-based microwave radiometers - IEEE transactions on geoscience and remote sensing, 2007

Janssen, M. A. (1994). Atmospheric remote sensing by microwave radiometry.

Küchler, N., D.D. Turner, U. Löhnert and S. Crewell, 2016: Calibrating ground-based microwave radiometers: Uncertainty and drifts. Radio Sci., 51 (4), 311-327. doi:10.1002/2015RS005826.

Maschwitz, G., U. Löhnert, S. Crewell, T., and D.D. Turner, 2013: Investigation of Ground-Based Microwave Radiometer Calibration Techniques at 530 hPa, Atmos. Meas. Tech., 6, 2641–2658, doi:10.5194/amt-6-2641-2013.

Meunier, V., U. Löhnert, P. Kollias, and S. Crewell, 2013: Biases caused by the Instrument Bandwidth and Beam Width on Simulated Brightness Temperature Measurements from Scanning Microwave Radiometers, Atmos. Meas. Tech. 6, 1171-1187, doi:10.5194/amt-6-1171-2013.

McGrath, A., & Hewison, T. (2001). Measuring the accuracy of MARSS—An airborne microwave radiometer. Journal of Atmospheric and Oceanic Technology, 18(12), 2003-2012

D V Land1, A P Levick2 and J W Hand, The use of the Allan deviation for the measurement of the noise and drift performance of microwave radiometers, 2007 IOP Publishing Ltd Measurement Science and Technology, Volume 18, Number 7

Vowinkel, B. (2013). Passive Mikrowellenradiometrie. Springer-Verlag

Wang, J. R., Racette, P. E., Piepmeier, J. R., Monosmith, B., & Manning, W. (2006). Airborne CoSMIR observations between 50 and 183 GHz over snow-covered Sierra Mountains. IEEE transactions on geoscience and remote sensing, 45(1), 55-61.

Woodhouse, I. H. (2017). Introduction to microwave remote sensing. CRC press.

---

## Referee Comment (RC2) · Anonymous Referee #2 · 31 Mar 2020

This paper describes analysis of the various components of an airborne passive microwave radiometer designed to estimate temperature profiles above and below an aircraft. Given that such an analysis has not previously been published, this work can serve as a valuable source of information for researchers attempting to use data acquired from the MTP. The paper's focus on instrument performance fits well within the scope of AMT subject matter.

While the analysis is thorough in that it considers the performance and uncertainty associated with individual components of the sensor as well as calibration methods used, some improvements to the paper are warranted. I recommend that the paper be

published with the following revisions.

General Comments

1. The paper sometimes reads more like a technical report than a journal article. I would suggest the authors begin with a broader view of such instruments, including their basic operating principles and their scientific applications. Reference to similar instruments should be included here as well. Then state the motivation for this work and how it supports research with MTP data.

2. The authors note that the MTP was developed by a team at JPL. While the developers have not published comprehensive instrument characteristics, one wonders whether they may have performed some of the work described in this paper. Have the authors reached out to the developers to understand whether this information exists within the JPL group, and if their results are consistent with the DLR team's findings?

3. While interesting, the work presented in Section 5 on sensitivity of LO frequencies and elevation angles seems to be outside the central theme of the paper. After presenting results on performance of various components, calibration methods, and associated uncertainties, it would seem more natural to discuss how performance and uncertainty impact the final measurement and applications. There is some reference to use of the data for gravity waves and the requisite accuracy for that application, but a more general discussion would make the paper more broadly relevant to readers.

4. Substantial improvement to the readability of the paper is needed. As noted in Comment 1 above, much of the information is presented as if this were a technical report. Following the Introduction, each section needs to begin with an overview of its contents, motivation for including that content, and how the content fits into the overall purpose of the paper. The material within a section is often not well-organized, paragraphs seem short and choppy, and transitions between topics are lacking.

Specific Comments and Questions

p4 eq. 2.1 - Is T the physical temperature? BT is defined here as brightness temperature, but elsewhere in the paper, TB is used (e.g., eq 3.3 on p9).

p4 line 28 - You state that the target is heated to a constant temperature of approximately 40C. In Table 1 the value is given as 41C. Why not just use 41C in both places?

p5 lines 1 - The explanation of brightness temperature is awkward and confusing. How about "...which is the temperature of an ideal blackbody emitting the equivalent radiance..."

p6 line 12 - Reference is made to the antenna diagram. It would be good to direct readers to the corresponding figure (Fig 2, I believe).

p6 line 13 - "half-sphere" should read "hemisphere"

p8 line 5-10 - It would be informative to share the range of ambient temperatures experienced outside the pod in flight.

p9 line 6 - "a" should read "at"

p10 line 16 - Section 4 includes uncertainty from pointing errors in addition to calibration methods. The title should reflect this, or the point error material should be placed elsewhere.

p14 line 23 - The sentence that begins with "Note that this definition of usable legs..." is confusing. I'm not sure what you mean.

p17 line 9 - This sentence lacks a verb.

p22 line 17 - If the authors choose to keep Section 5 as a discussion of new measurement strategy, it would be interesting to demonstrate the impact of LO shifts and/or elevation angle changes on simulated data.

p22 line 31 - "full-with-half-maximum" should read "full-width-half-maximum"

p34 Figure 5 - A legend is needed here

p40 Figure 11 and 12 - These figures are too small to differentiate the individual lines/methods.

p44 Figure 15 (left panel) - It's impossible to distinguish the 58.363 GHz line from the 56.363 GHz line

p45 Figure 16 - The legend indicates lines for 6 altitudes are shown, but I can only see 4 on the left plot.

---

## Author Comment (AC1) · 26 May 2020

AMT-2019-455

**Measurement Characteristics of an airborne Microwave Temperature Profiler (MTP)**

by Mareike Kenntner et al.

Reply to the referee #1 comments

We would like to thank the referee for thoroughly reviewing our manuscript and for the helpful advice provided in the comments, below, which we believe helped improving the manuscript. In the following we give our answers regarding the points made by the reviewer. The statements, comments and suggested corrections raised by the referees are printed in black italics and our comments are presented in blue. We tried to consider all of the raised points in the revised manuscript in an adequate manner.

Answers to referee #1

*General Comments:*

*1. The implications of the instrument characterisation for the subsequent interpretation of TB and temperature retrieval are not thoroughly assessed. Section 5 should have a clear outcome on the questions: (i) spectral characteristics: Which are the representative frequencies of the three channels? Which frequencies shall be assumed for the retrieval algorithm? Does the RT have to consider the full bandpass characteristics? (ii) Which is the effect of the antenna bandwidth? Is a pencil beam approach justified? (III) What noise characteristics have to be assumed in the retrieval, e.g. in the measurement covariance matrix?*
This is a very thorough list of characteristics to be assessed for a retrieval algorithm. Currently, there exists more than one approach to retrieve the absolute temperature profiles from MTP brightness temperatures, which is one of the reasons why the authors decided to not consider retrieval algorithms in the study.

The intention of Section 5 was to point out that a different approach to decide which measurement strategy (i.e. number of viewing angles and LO frequencies) could be used. A thorough study of all implications for a retrieval algorithm would be beyond the scope of this study. For that reason and on advice of the reviewer, Section 5 has now been moved to an appendix. The usefulness of an assessment of the questions asked above, as well as to investigate the impact of changes in the measurement strategy are understood, and should be attempted in an upcoming study.

*2. Accurate calibration is the most important task in microwave radiometry. As all components are strongly temperature dependent besides temperature stabilisation a periodic calibration is needed. The calibration might only update the gain of the system (relative calibration) or make an absolute calibration in which all parameters of the raw measurement (count) to TB model are derived. In the simple linear case (as it is used in this manuscript) these are gain and receiver noise temperature Tr which can be derived by pointing the antenna successively to two reference targets. The authors seem to be not aware of this classical microwave formalism which is also apparent as they hardly cite any literature microwave radiometry (list in the back) and some flaws in the radiometer formula application. The major questions which would need to be ad dressed are: How good are the reference targets (blackbodies)? How frequently does a calibration need to be made? Why have the measurements in the cold chamber with view on a stable target not been used for such an analysis? The next step would then be the in flight calibration. Assuming that the laboratory*

*calibration (strategy 1) would work is a bit naïve. However, there are good approaches later on using the horizontally pointing measurements but a motivation and explanation why this procedure was chosen needs to come first.*

We think it is not necessary to explain the standard method of radiometer calibration, here. The presented assessment of calibration methods is based on using this calibration method, since it was used throughout the measurement series inside the cold chamber, and the derived calibration parameters are used to characterise various parts of the instrument, and calibration methods used in flight. As pointed out within the manuscript, the instrument only has one single built-in calibration target, which can be used during flight, and the study provides a guideline, how this can be used to derive the brightness temperatures from the measurements taken during flight.

As for the specific questions asked by the reviewer, we have added specific information about the microwave absorber used in the laboratory calibration, as well as some information from the instrument documentation.

Concerning an analysis of the necessary frequency of calibration, please see our comment on the specific comment about an assessment using the Allan Variance, below. In general, the deployment of the MTP instrument, which is mounted completely outside of the aircraft, requires permanent monitoring of the instrument state, which is already implemented in the way the measurements are taken: Calibration measurements are made after each measurement cycle (i.e. after taking measurements at each elevation angle), approx. every 13-14 seconds. This is the highest possible frequency with which calibration measurements can be taken during flight.

Finally, while the proposed strategy no. 1 for in-flight calibration might sound naïve, we have included this approach in our discussion for reasons of completeness. The results show that using this method in the cold-chamber provides the possibility to link in-flight instrument conditions to measurements within the cold-chamber, and derive calibration parameters, that lead to comparable results as other calibration methods.

In the revised manuscript we therefore have included a brief discussion of the pros and cons of using each of the calibration strategies when introducing them.

3. *The information on the MTP measurement principle is not clearly provided in the beginning of the manuscript making it difficult for the reader to follow. Bits and pieces come together at different instances, e.g. scanning is explained on page 14 and especially the discussion on the use of different oxygen lines is confusing. For better understanding the authors should include a thorough description of the MTP measurement principle in the beginning and add an absorption spectrum (preferably even for different pressure levels as in Fig. 16) to illustrate the frequency channels (and their potential tuning range). This also serves to introduce the double sideband principle. Further, it could be explained why the LO is typically set at center frequency for mitigating problems due to frequency drifts, and how non-resonant emission (water vapor continuum, hydrometeors) affects the measurement. This would also demonstrate that the LO frequency is not the frequency for which the measured TB is representative (passband averaged – see Fig. 16).*

The revised manuscript now contains more information on how the measurement set-up works, and links to the already existing literature, in which the standard settings for this instrument are already introduced and discussed (i.e. Mahoney and Denning, 2009, and Lim et al., 2013). We have updated Table 1 to include all relevant information.

The measurement principle is explained in more detail in the revised version.

*4. Section 5 address future measurement strategies in terms of frequency selection and elevation scanning. This is an important study but is not done as thoroughly as it is needed especially in light of vertical resolution of the retrieved temperature profiles for different types of atmosphere. It also does not take into account the findings of their laboratory measurements in respect to the spectral and spatial sensitivities. As the paper is already very lengthy it should be taken out.*

We agree that more could be done to thoroughly assess all input parameters needed in a retrieval. We are aware that those input parameters may depend on the type of retrieval used (optimal estimation, neuronal network, Tikhonov retrieval,…), and the Section was intended to indicate potential for possible improvements in the general measurement strategy. While we do think that this consideration is worthy to be noted by users of the MTP data, and other groups using MTP instruments, we agree, that it might better fit as an appendix.

*5. At several instances it seems that the authors have gravity wave detection as application in their mind – this is ok but needs to be clearly stated (only abstract). Many readers might not know which requirements in TB are needed for this purpose. Other users might be more interested in vertical resolution for stability assessment.*

Indeed, the study was first undertaken with the goal of assessing temperature fluctuations for gravity wave studies. As the short overview of already published studies using MTP data shows, there are other interests in using this kind of data, as well. We have changed the text in the abstract and introduction accordingly.

*6. The readability of the paper needs to be improved - sometimes it is more a technical report than a paper. No clear goals are provided, the structure is not always clear, the text is written rather lengthy and many basic informations only appear rather late in a middle of a section where you would not expect it. Short paragraphs sometimes even only one sentence long occur and the text frequently repeats (unnecessarily) the figure captions, e.g. "Plotted is also a..". The paper could be shortened by reducing number of figures or using an appendix. I would recommend to concentrate only on the past measurements. The optimized scanning strategy In case but the future – which I think would be an own study if done carefully could go in an appendix.*

As noted above, we agree, that section 5 fits better as an appendix to the study, and have changed the manuscript accordingly. The wording of the manuscript has been checked, and where necessary, revised.

Figure 8 in the revised version now contains Figures 8 and 10 of the submitted version, and Figure 10 of the revised manuscript now contains both Figures 11 and 12 of the submitted version.

*Specific Comments: Why are brightness temperatures referred to as BT in the text (and Fig. 11) and TB in the equations. Historically the satellite community uses BT and the ground-based community TB. I don't think it matters which one is chosen but it should be consistent.*

All instances of "BT" in the text have been changed to "TB" for consistency.

*P1l8: "records radiances", no it records counts which are calibrated to brightness temperatures - it is ok to say TB here*

Indeed, the recorded signal is in counts. Since the physical quantity that is measured is radiance, the sentence is changed to: "The Microwave Temperature Profiler (MTP), an airborne passive microwave radiometer, measures radiances, recorded as counts and calibrated to brightness temperatures, in order to estimate temperature profiles around flight altitude."

*P1l9: "state of the atmosphere can be derived" this indicates much more information than the temperature profile which was stated already – what else?*

The (dynamical) state of the atmosphere can be derived from the temperature profile; The sentence is extended to: "From these data quantities such as potential temperature gradients and static stability, indicating the state of the atmosphere, can be derived and used to assess important dynamical processes (e.g. gravity waves or stability assessments)."

*P1l22: "weaker oxygen lines" better write 'frequency channel'. The LO frequency of the channel does not necessarily need to be at a line center. Also and it seems to me that it is not clear to authors: the LO frequency is not the representative frequency of the channel – and the "representative frequency" can be extracted from their laboratory measurements. I anyway suggest to modify section 5 such that it can provide the necessary input for the retrieval algorithm*

In the case of the MTP, the LO frequency really is placed at the line centre, which is stated in section 2.2. The discussion of the "representative frequency relates to the radiative transfer calculation, which is part of the retrieval algorithm. The transmission function was measured for the MTP, and should be used when setting up the retrieval calculations. In section 5, a simplified approach was chosen, to show that some easy-to-be-made changes to the measurement set-up can already have a large influence on the quality of retrieval input. Since this section will be shifted to the appendix, we will not attempt to make additions, as a thorough investigation of implications for radiative calculations may well fill its own study.

*P1l22: "calibration parameters do clearly depend on the state of the instrument". This is the key in microwave radiometry for astronomy, atmospheric, planetary science etc. ever since and for all instruments there is the question how frequently one has to calibrate, e.g. Dicke switching for short-term fluctuations. Unfortunately, even slight vibrations and temperature changes can cause transmission characteristics to change thus calibration parameters. So this sounds a bit naive – I recommend the authors to look more in basic microwave radiometry books, e.g. Janssen, 1994, Vowinkel, 2013, Woodhouse, 2017,*

We agree that this problem is not unique to the MTP. Microwave instruments always have this issue, and the literature states many examples and approaches used to stabilize the systems in operation. Since the MTP is mounted completely outside an aircraft, which frequently changes flight altitude and/or speed, and enters different temperature and wind conditions, such stabilization is not readily possible. Moreover, changing the instrument design in an attempt to increase instrument stability would be very costly, and the limited space inside the wing canister does not allow for many options in hardware changes anyway. This increases the problem for the MTP, as compared to, e.g., ground-based instruments. We have changed the sentence to show these circumstances: "The MTP shows quite large changes of the instrument state, imposing considerable changes in calibration parameters over the course of a single measurement flight"

*P1l26: Here it should be said that precision is determined for TB which closely relates to the atmospheric temperature when the instrument is pointed horizontally – otherwise it is confusing*

We have made the suggested change.

*P2l16. What is meant by structures?*

Basically it is meant that any signal in the timeline of measurements, that deviates from a smooth background could be caused by either noise created by the instrument itself, or has its cause in some real, physical process in the atmosphere, in which case we would like to be able to detect it with this instrument.
The word "structures" was replaced by "fluctuations".

*P3l9-12 and P3l14: There is a very long list of applications of past studies using older versions of the MTP (is that really necessary?) and then it is claimed that instrument characteristics need to be known for correct interpretation. This is true and that's why this study is valid but it somehow implies that the work here also helps with data from old campaigns. This needs to be clarified.*

It was not the authors' intention to claim that this study helps interpreting measurements from older MTP instruments. Listing the previous usage of MTP data from the past was intended to show that data from MTP instruments developed by JPL is frequently used and that this study is of interest to a wider community of researchers.

The paragraph has been reworded so that there is no impression given that older studies may not be valid due to the herein presented findings. In the manuscript we only state that thorough characterisations of older MTP instruments have not been published before. We are clearly not implying that those characteristics were always completely unknown!

*Introduction: the whole introduction is dedicated to the MTP but there is no reference to other studies on the characterisation of other microwave airborne instruments is made, e.g. Blackwell et al, 2001 describing NAST with frequencies 50-57 GHz, McGrawth and Hewison, 2001, Wang et al, 2007 etc. which might also check different instrumental parameters. The introduction clearly needs to mention the goals of the lab investigations.*

The studies mentioned by the reviewer provide a variety of instrument characteristics, many of which were also referred to here. Other characteristics measured, e.g. in McGrawth and Hewison were not an option for the DLR.MTP, as a disassembly of the instrument hardware was not possible without losing aircraft certification. This is now stated in the introduction and in the appropriate sections, in connection with a reference, where fitting.

*P4l2: Not all radiometers for temperature profiling measure at the oxygen absorption complex around 60 GHz - also 118 GHz is used. In general, it is surprising that no reference is made to the fact that operational meteorological satellite instruments, e.g. AMSU-A, do temperature sounding since decades. These sounders exploit only the frequency information for profiling while the MTP aims at improving the resolution by angular information. It is necessary to explain the measurement principle here thoroughly, showing a spectrum (ideally for different altitudes) and the considered frequency channels. On a side note: The accuracy of the oxygen spectroscopy is still under debate which is, however, more important for retrievals, Caddedu et al, 2007; Cimini et al, 2018, Maschwitz et al 2013.*

It is true, that this study places its focus on the MTP instrument itself. We have changed the wording to acknowledge the fact that temperature measurements are also possible at 118 GHz. The comment on using the angular information to improve the vertical resolution is very valuable, and we changed the text in Section 2.1 accordingly.

The development of the MTP instrument has not been done by the authors of this study. Since the instrument has a long history, there are a number of publications available, explaining the measurement principle (e.g. Denning, et al. 1989), as well as some unique features and considerations related to the wing-canister design (e.g. Lim et al. 2013), including consideration of the used frequencies. We have given those references in the description of the MTP instrument (Section 2.2) and added a few sentences to briefly introduce the measurement principle.

Concerning the very interesting ongoing debate of the accuracy of oxygen spectroscopy, we agree that this topic relates more to retrievals, which has been explicitly excluded from this study.

*P4l2: Why don't you explain the heterodyne principle and talk about a double side band receiver. This is very important to clearly define the frequencies for the radiative transfer used for retrieval development.*

The important information that the DLR-MTP uses a double-sideband heterodyne receiver is added in the text, and the measurement principle is briefly explained in section 2 of the revised manuscript.

*P4l18: "making the retrieval of temperature profiles possible" Most instruments only use information on frequency dependence. Make clear that the MTP can achieve higher vertical resolution by adding the angular information.*

We agree, that it is valuable to pointing out a clear advantage of the MTP instrument in comparison to other microwave systems. We have revised the text, making this point clearer (also according to the comment referring to p4l2).

*P4l24: Thermal stabilisation is the most important part in a microwave radiometer the performance of all microwave components strongly depends on temperature. Therefore more details on that are needed.*

Details on the temperature control are given in Mahoney and Denning, 2009, to which we refer at the beginning of the Section. According to this publication the "[t]emperature control at the point where the thermistor is mounted is approximately ±0.1C".

There is no possibility to monitor the real temperatures of the components during flight, other than through the housekeeping data, recorded during flight. Here, we do see changes depending on flight levels (surrounding temperatures), and temperature gradients across the instrument are visible. However, since the thermistors are only placed at certain positions, there remains the question, if the temperature recorded in the MTP housekeeping data is representative for the critical components, and how changes are to be interpreted.

We do acknowledge that a new series of cold-chamber laboratory measurements to investigate the overall temperature behaviour of MTP components could be an interesting

study in the future, but such measurements cannot be performed in the near future to be included in this study.

*P4l229: What about temperature stability, homogeneity, spill over of the target, cf. Mc Grawth and Hewison (2001).*

While we were not able to perform all of the characterisations described in the study published by McGrawth and Hewison, most tests were also performed in our laboratory set-up.

The main reason for leaving out some of the tests, e.g. determining the spill-over, is due to the fact, that the DLR-MTP is certified to be flown on a research aircraft. Disassembling the instrument would lead to a costly process of re-certification, which prevented us from taking any parts out of the instrument. Hence, a spill-over measurement for the antenna was not possible. For the same reason, we do not have the means to add any observation system to the instrument while mounted on the aircraft, to, e.g. monitor the thermal stability during flight, or the characteristic temperature of the heated target.

The temperature stability of the target is part of the investigation presented in Section 4. The housekeeping data only state the thermistor temperature in the back of the target, other measurements are not available. Testing the representative brightness temperature of the hot target in the cold-chamber is the nearest we can cat to such an assessment. There is no possibility to conduct an assessment comparable to that shown in McGrath and Hweison, 2001, since we cannot change or add parts to the hardware of the instrument, without a costly revision of the permit-to-fly.

*P5, l14-15: the discussion on the oxygen spectrum and LO needs further explanation and should come before not in the section on wing-canister, same for the information on the frequency range (l25) below.*

The Section is structured in a way to introduce the basic principle of Microwave radiometry, moving from the broad principle to more and more specific details.

As mentioned in a previous answer, the discussion of the Oxygen spectrum and choice of LO is already presented in Mahoney and Denning, 2009, and also briefly discussed on Lim et al., 2013. We have added a sentence to the text, which points out that the original choice of standards LO frequencies was made by the inventors of the instrument.

*P5l22: how large is the gap, x MHz?*
The gap is nominally 20 MHz wide, which is confirmed by the measurement of the filter function (Figure 2, left panel). We have added this information in the text, as well as a reference to Figure 2.

*P6l 6 "investigation OF calibration"*

We have corrected this typo.

*Section 3. The frequency response of the bandpass is investigated but there is no discussion on the stability of the LO frequency – does this have any potential effect on measured TB?*

A measurement of LO frequency stability would require disassembling part of the instrument, to attach the measurement equipment (oscillator) to the frequency synthesizer output. We did not attempt such measurements, which required disassembling the instrument, as this would have had serious implications for continued airworthiness.

Some thoughts on the potential influence of measured brightness temperature, influenced by synthesizer errors: Potentially the largest influence is that with a shift of the LO frequency, the gap in the middle of the filter function is no longer located at the center of the strong oxygen absorption line. Hence, on one side of the filter function, much larger absorption very close to the aircraft would be included in the signal, while on the other side, the signal is caused by absorption slightly further away from the aircraft. Since the LOs are placed at very strong absorption lines, the first affect probably outweighs the second effect, so that the measured brightness temperature would be representative of an altitude layer closer to the aircraft, than assumed. The absolute error depends on the aircraft altitude and the temperature gradient present in the atmosphere surrounding the aircraft. Largest errors would certainly be induced at lower pressure, where the line shape is sharper. Also, larger temperature gradients in the atmosphere would induce a larger error in the measured brightness temperature. The absolute effect of frequency shifts in the LO would have to be modelled, using radiative transfer calculations. However, we feel that the large gain fluctuations seen in the calibration of campaign data can be assumed to be much larger than the induced error by small LO frequency shifts.

Following the comment from Reviewer 2, that the pointing error seems misplaced in the Section on brightness temperature calibration, we have added a new Section to discuss further sources of measurement uncertainty, in which the pointing error is included, as well as some discussion of other error sources, such as synthesizer errors, reflecting the discussion above.

*P6l27: The authors mention the periodicity of the signal first. I understand that for gravity wave detection this is important but in terms of radiometer performance the most important question is whether the instrument follows the radiometer formulae (Eq. 4.8), i.e. noise reduces with increasing integration time. For this purpose typically the Allan variance is used. This characterizes the noise and determines how long measurements can be integrated in time and how frequently a calibration needs to be performed.*

We indeed measured time series of the Tb in the cold chamber. It was found out that up to averaging times of > 20 s the noise behaviour resembles white noise. The precision of the measurement could thus be improved by longer averaging times, however, the spatial resolution due to the high aircraft speed clearly calls for reducing the measurement cycle to short integration times.

A brief discussion of those points is added in the revised manuscript.

*Section 3.1: The name is irritating as it could mean much more. The measurements of the bandpass characteristics and the antenna diagram (section 3.1) are important and interesting but are presented rather briefly without any implications for the subsequent retrieval. Even the exact measured bandwidth and beamwidth are not given. For the analysis or implications RT calculation would play a major role. As for example shown in Crewell et al. (2012, their figure 10) the bandpass characteristics can cause the effectively measured TB being representative for a frequency deviating significantly from the specified channel frequency. In fact in the double side band approach this anyway takes place and needs to be handled in the RT underlying the retrieval process. Similarly, the antenna pattern smears out*

*atmospheric features especially at low deviations from the horizontal in a vertically stratified atmosphere (Meunier et al., 2013). To appreciate this laboratory measurements and their impact on the measured TB further analysis is required which would fit well into section 5.*

In Crewell, 2010, the authors state that ".Because of the strong nonlinear changes in brightness temperature with frequency when atmospheric spectral features are measured the detector's exact band-pass characteristics have to be taken into account". Our study presents exactly this transmission function, needed for the correct set-up of a retrieval algorithm. Similarly, the antenna pattern is presented in this study, and should, ideally, be used in any retrieval algorithm used to derive the atmospheric temperature profiles.

We did explicitly state that we do not discuss retrieval methods and associated uncertainties; Hence, we feel that while those are important points to consider in a retrieval set-up, a thorough discussion of these effects would go beyond the scope of this study.

*P7l12: "A certain 'waviness' is visible next to this" ripples are typical in any microwave component due to EM wave theory propagation – reducing the amplitude is key.*

Thank you for your comment on this observation, clarifying the source of the observed signal. Since the structure is now known through the measurement of the transmission function, it can be considered in RT calculations. Attempting to reduce the amplitude would necessitate the disassembling of the instrument, and possibly replacing parts, which, as mentioned before, has serious implications for aircraft certification.

*P7l23: how stable is the noise diode, how much does it depend on temperature (stabilization)?*

There is an entire section (4.2) dedicated to this question. We have added a reference to this section. When investigating measurement flight data, it is mentioned, that on top of general noise diode signal dependence on temperature, we did experience technical problems due to a cold soldering joint in our measurement campaign, so it is not possible to make statements about the stability in real flight conditions.

*P8l14: "takes some time to stabilize".. needs to be more quantitative – later it is mentioned but not here*

Actually, quantification is hard due to the fact that this stabilization depends a lot on the environment, and the way the instrument is operated. We did observe quite different times the instrument took to stabilise – between different operating environments. We have added a sentence in the revised text to acknowledge this fact.

*Section 3.2: Information on the accuracy of the target temperatures is missing. P9l14 mentions the "hot" target – should be explained before*

The reference "hot target" is introduced at the beginning of Section 3.2, in the paragraph that describes the laboratory settings, and which targets are used. (p.8, l21 in the discussion paper).

The heating of the target is done in the same way as the heating of other parts of the instrument (see Mahoney and Denning, 2009). The thermistors are heating the components to within an accuracy of +/- 0.1C (private communication with Mahoney and Denning)

The temperature actually seen in the calibration process is investigated in Section 4.2. Figure 10 shows a constant reading of the "target temperature" from the MTP housekeeping data throughout the entire measurement series in the cold-chamber. There is no other measurement of the target temperature available, and external monitoring, e.g. through an infrared camera, cannot be realised in flight, due to the aircraft certification process. (See also our answer to your comment referring to p4/l29).

*P8l30: I find the term "at all LOs" confusing – also at other instances. Why not write for all frequency channels?*

Given that the use of "LO" might be misleading to some readers, we gladly follow this suggestion, to rather use the wording "all frequency channels".

*P9l7: Why do the authors not use the classical microwave notation using the gain (cf, Janzen, Mc Grawth and Hewison, ? The difference between receiver and system noise temperature needs to be made clear.*

We chose this approach based on the investigation of "how to best calculate the brightness temperature from a known recorded signal (counts)", i.e. looking for a way to calculate the brightness temperature as a function of the recorded counts, contrary to the traditional approach taken in microwave radiometry. However, this approach is still similar in the way that the defined slope of the line ("S_cal") in this study, is the inverse of the traditional definition of the Gain, while the receiver noise temperature is still defined in the same way, as in the classical formulation of microwave radiometry.

Since the classical notation is much better known, the authors do acknowledge this fact, by adding a note in the revised manuscript.

*P9l17: Radiometers are never completely stable which is why periodic calibrations have to be made. In between this calibrations the TB could be corrected assuming a linear trend as shown in Fig. 6. The following paragraph describes this for the airborne measurements bit it is unclear for me that for these linear fits segments of 5 min without calibration are used?*

The 5 minutes mentioned in the following paragraph do not refer to the time between two calibration measurements, but to the length of the flight segment from which data is being used. Calibration measurements during flight are part of every single measurement cycle (i.e. one calibration measurement every 13-14 seconds!). We have added this information in the revised manuscript.

*P10l1 and following: The spectral analysis is interesting and similar to the Allan variance but is unclear to me why it is applied to atmospheric measurements and not to the cold chamber measurements where the real instrument performance could be tested. The concatenation eliminates real temporal signals. Does the analysis differ between in flight and laboratory measurements .*

We used the flight segments, because some of those are much longer than the cold-chamber measurement segments. When comparing the analysis from the flight segments to drift measurements in the laboratory, the results do look similar, as long as the drift period is included. If it is not included, the result from the drift measurements indicate a smaller correlation coefficient alpha, showing that without drift, the instrument noise behaves more like white noise.

Using the campaign measurements has the advantage, that the parameters used to test significance in the data analysis represent much more conservative limitations, so that the confidence in the results is higher.

*P10l20-27: "line parameters" is irritating as it could be interpreted in spectral lines: it is about the updating your calibration model, basically, gain and receiver noise temperature. It looks like the authors are not too familiar with typical microwave calibration techniques which is reflected by the lack of citation of microwave radiometer basics and studies. In operational receivers many strategies for that exist (Maschwitz et al., 2013) as typically gain needs to be adjusted more frequently than TR, relative/absolute calibration.*

In the revised version we only use the term "calibration parameters". As mentioned in some answers above, the calibration measurements are performed as frequently as possible during deployments.

*P11lEquation: Why so complicated $T_r^{CCh}(C\_hot)$ and not simply $T_r$ – explain the meaning of the different indices.*

$T\_r^{CCh}$ is derived using different measurements than $T\_r^{ND}$, since they are calculated using different calibration methods, which is shown by using the indices. It is now explained in the text.

*P11l19: Give values to underline the statement*
This comment refers to the statement that when applying the calibration method that is based on the hot target and the noise diode "[…] two reference temperatures are used, which are above the expected measurement range.

As suggested by Reviewer two, we have added a sentence in the beginning of the document, that the atmospheric temperatures surrounding the aircraft (and therefore measured by the MTP), are within a range of 190K - 260K in flight. Higher temperatures – up to 300 K are also possible at very low flight levels. As explained in the instrument description, the hot target has a temperature control keeping its back side at a temperature of 45C (just below 320K), and the noise diode signal is added to this temperature, which is mentioned in this very paragraph. Hence, with the addition of expected atmospheric temperatures, it should now be clear that both temperatures used for calibration are (well) above the expected atmospheric temperatures measured by the MTP. A sentence is added in the revised text, to underline the statement.

*P12l4: The calibration strategies might serve different purposes. That the first strategy leads to comparable results seems astonishing.*
Finding that laboratory values can be used to calibrate flight data is, indeed, astonishing. However, the laboratory data we refer to were produced in very specific conditions, meant to imitate flight conditions. Those results cannot be achieved by a single calibration on the ground, since the trends in changing calibration parameters cannot be reproduced without the specific settings used in our laboratory set-up. It was the purpose of those specific settings to mimic flight conditions as well as possible, and the results show that of all changing parameters, that can influence the instrument during flight, the temperature has the most important influence. This is in agreement with the finding of McGrath and Hewison, 2001, who also found the largest dependence of instrument parameters on temperature changes.

*P12l12: The cause for the standing waves is the refractive index of the LN2 – here Küchler et al., 2016 should be cited for details. Here it sounds that just the evaporation is the reason*
We have revised the sentence, so that it now should be clear that the _changing_ interference with the standing wave is caused by the evaporation of the liquid nitrogen. A reference to Section 4.1.1 in Küchler et al., 2016 has also been added, since it gives useful background to the standing wave problem.

*P12l25: Of course the calibration parameters change with changing environmental conditions if the temperature stabilization of the instrument is not perfect. The question to ask if this is repeatable. Would the same parameters be measured if the instrument had been moved and electrically disconnected in between?*
Single calibration measurements, using the hot-cold method were performed before and after a number of campaign deployments, but did not show any consistent picture, due to the large influence of the surrounding temperatures on the MTP instrument. These surroundings cannot be influenced while the MTP is mounted on the aircraft, hence, it is not easy to repeat the measurements needed to establish this consistence. However, all calibration parameters derived during those hot-cold calibration during campaigns were within the range of observed calibration parameters during the cold-chamber measurement series, which is a strong indication that un-mounting the instrument and installing it in the cold-chamber did not have significant impact on the parameters. This is a very strong indication that repeatability is given between different MTP campaign deployments.

*P13l29: Why is the temperature unknown – more discussion is needed – see Mc Grawth and Hewison, 2001.*
We have mentioned the temperature gradient between the back of the target (which is heated), and the (not heated) front of the absorber, to which the measurement is most sensitive to. We have re-worded the paragraph, so that this fact becomes clearer.

*P16l8: why do you explain this only here and not at the beginning of the calibration section*
The order of the section of uncertainty estimation has been changed accordingly in the revised manuscript.

*P16l13: Nobody remembers counts better give the atmospheric temperatures and notate the counts with c_min and c_mac or later c_ref instead of 18500.*
The notation follows the calculation. For better understandability, we have added the corresponding temperatures.

*P16l24: "The vertical, grey shaded.." this is not paper style. The figure should be only a reference for the text.*
We have moved the descriptive part of the text to the figure caption.

*P17l9: "In literature" then give a reference*
The reference to Ulaby, Moore, and Fung, 1981, and Woodhouse, 2005 were already given after the equation was stated. We have moved this reference to the beginning of the sentence.

*P17l9 to 29: This paragraph shows that the authors have not much experience with microwave radiometry. It is weird to present the well established radiometer formula at the end and not in the beginning. The formula describes the internal noise of an ideal radiometer and typically one just writes a proportionality and not an equal sign as other losses occur (e.g. factor 2 for Dicke switching). Further, the authors put in 400 MHz as bandwidth but the double sideband receiver only has 200 MHz in the IF. The most important think to look at the*

*radiometer formula is to check if the noise decreases with longer integration time which is basically what the Allan Variance technique does – it finds out at which point gain fluctuations dominate. This should be checked by the laboratory measurements in the beginning and not in this section. Note, it is strange to only now to provide the integration time for atmospheric measurements.*

We have switched the order of the paragraphs, so that it now starts with the theoretical formula. It is correct, that only the 200 MHz bandwith is to be used, which has been corrected. The new uncertainty value is 0.117 K; assuming an atmospheric temperature of 250 (0.108 K for T_atmo = 190 K, and 0.1255 K for T_atmo = 300 K). This is still considerably smaller than the error derived from calibration parameters, so the overall message remains true.
The integration time used for recording measurements was added in the table.

*P18l14: If the dominant uncertainty is the noise couldn't it be reduced by longer integration times?*
There are two possible ways to reduce the noise: 1) to change parts of the instrument, which is not possible due to the aircraft certification. The other possibility is to increase the integration time while recording the signal. This implies a longer recording time for a single measurement cycle, which in turn reduces the horizontal resolution of the measurement, due to the high speed of the jet-engine aircraft the DLR-MTP is flown on. The current settings are recommended by the inventors of the instrument, who have already considered the best compromise between instrument noise figure and measurement resolution (private communication). We have added a sentence to remind the reader of the fact that noise reduction is only available at the expense of horizontal resolution.

*P18l30: LO frequency*

"frequency" was added in the text.

*Table 1 does not include all instrument characteristics of interest, e.g. receiver noise temperatures, integration time, polarization. I am missing information on microwave window transmission*

The table has been updated.

*Fig. 8 could be combined with Fig. 10*

We have combined the two figures in the revised version.

*Fig 11: Different calibrations need to be explained in figure caption. Caption does not say how the difference is calculated (what is the reference – the overall mean?). As I do not see significant temporal development mean and standard deviation could be just added as last lines in Table 6.*

The reference of the shown brightness temperature difference is the time series of brightness temperatures derived with method TTS1. We have updated the figure caption to include a reference to Table 6, which explains the calibration methods.

---

## Author Comment (AC2) · 26 May 2020

AMT-2019-455

**Measurement Characteristics of an airborne Microwave Temperature Profiler (MTP)**

by Mareike Kenntner et al.

Reply to the referee #2 comments

We like to thank the reviewer for providing helpful advice to improve the quality of our study. In the following we give our reply regarding the points raised by the reviewer #2. The statements and comments given by the referee are printed in black italics and our comments are presented in blue.

Answers to referee #2

General Comments

*The paper sometimes reads more like a technical report than a journal article. I would suggest the authors begin with a broader view of such instruments, including their basic operating principles and their scientific applications. Reference to similar instruments should be included here as well. Then state the motivation for this work and how it supports research with MTP data.*

The intent of our study is to provide the specific characteristics of this particular design of MTP instrument, purchased from JPL. We would like to remind the reviewer that this instrument was not designed by the authors of this study, and this publication is not intended to be a general introduction to the instrument. Such an introduction was already given in the studies mentioned in our publication (e.g. Mahoney and Denning, 2009; Lim et al., 2013), and as a result, we do not see the necessity to include the suggested overview of microwave instruments in use.

*The authors note that the MTP was developed by a team at JPL. While the developers have not published comprehensive instrument characteristics, one wonders whether they may have performed some of the work described in this paper. Have the authors reached out to the developers to understand whether this information exists within the JPL group, and if their results are consistent with the DLR team's findings?*

Indeed, we did have contact to the group at JPL. The instrument design and the components used in this design, differ from those used in earlier designs of the instrument. Earlier versions of the instrument are mounted inside the aircraft cabin, and hence, the investigation of changing surrounding temperatures has not been performed by JPL. Some results and observed instrument behaviour and characteristics were partly discussed with JPL staff, and regarded as expectable. They also compare well to figures and characteristics shown in older documentation, (e.g. the patent for the horn antenna and rotating mirror design from 1976; Fletcher et al.; United States Patent for a Highly Efficient Antenna System Using a Corrugated Horn and Scanning Hyperbolic Reflector; US Patent No. 3,949,404).and the considerations about the instrument design; given in the internal documents/ private communication with JPL provided with the transfer of the instrument to DLR.

*While interesting, the work presented in Section 5 on sensitivity of LO frequencies and elevation angles seems to be outside the central theme of the paper. After presenting results on performance of various components, calibration methods, and associated uncertainties, it would seem more natural to discuss how performance and uncertainty impact the final measurement and applications. There is some reference to use of the data for gravity waves*

*and the requisite accuracy for that application, but a more general discussion would make the paper more broadly relevant to readers.*

We agree with both reviewers, that Section 5 of the paper fits better as an appendix, as it only presents a very brief investigation of fast-to-apply improvements to the measurement strategy. However, as noted by reviewer 1, there is a lot of room for interesting investigations into implications for radiative transfer calculations and retrieval error estimation, which, if attempted, should be presented in a study of its own. We decided to move this section to an appendix. The in-depth assessment of measurement strategy impacts on the retrieved temperature profiles, including the general discussion about consequences for data analysis, should be considered in a study of its own.

*Substantial improvement to the readability of the paper is needed. As noted in Comment 1 above, much of the information is presented as if this were a technical report. Following the Introduction, each section needs to begin with an overview of its contents, motivation for including that content, and how the content fits into the overall purpose of the paper. The material within a section is often not well-organized, paragraphs seem short and choppy, and transitions between topics are lacking.*

We have revised the manuscript according to the reviewer's recommendation.

Specific Comments and Questions

*P4 eq. 2.1 - Is T the physical temperature? BT is defined here as brightness temperature, but elsewhere in the paper, TB is used (e.g., eq 3.3 on p9).*

Yes, this is the Planck equation, based on absolute temperature T.

Following a comment from reviewer #1, we have changed all instances of "BT" in the text to "TB" to be more consistent in the use of the abbreviation of brightness temperature.

*p4 line 28 - You state that the target is heated to a constant temperature of approximately 40C. In Table 1 the value is given as 41C. Why not just use 41C in both places?*

Looking back at our data, we have corrected both values to 45C, which better represents the temperature of the thermostats used to heat the target (recorded in the MTP's housekeeping data, and shown as orange line in Fig. 8 of the revised manuscript).

*p5 lines 1 - The explanation of brightness temperature is awkward and confusing. How about "...which is the temperature of an ideal blackbody emitting the equivalent radiance..."*

We have changed the wording according to the reviewer's recommendation Thank you for this helpful suggestion.

*p6 line 12 - Reference is made to the antenna diagram. It would be good to direct readers to the corresponding figure (Fig 2, I believe).*

We have added the reference to figure 2, both for the antenna diagram, as well as for the earlier mentioned instrument transmission function.

*p6 line 13 - "half-sphere" should read "hemisphere"*

We have corrected the wording of the revised text.

*p8 line 5-10 - It would be informative to share the range of ambient temperatures experienced outside the pod in flight.*

We agree. The information (190 – 260K, or warmer, if lower flight levels are flown – up to 300K at the surface ) is added in the revised text.

*p9 line 6 - "a" should read "at"*

The typo is corrected in the revised version of the manuscript.

*p10 line 16 - Section 4 includes uncertainty from pointing errors in addition to calibration methods. The title should reflect this, or the point error material should be placed elsewhere.*

As suggested earlier, a broader discussion of implications for brightness temperature error is desirable, and some other characteristics influencing the measurement accuracy could also be discussed, as pointed out by Reviewer one. Hence, we have added a new Section (Section 5 in the revised manuscript) to discuss further sources of measurement uncertainty, in which the pointing error is included, as well as some discussion of other error sources, such as synthesizer errors.

*p14 line 23 - The sentence that begins with "Note that this definition of usable legs..." is confusing. I'm not sure what you mean.*

The chosen flight segments are cut so that no flight manoeuvres lead to large changes in the measured signals. The cutting criteria are based on aircraft parameters only – not on any readings of MTP housekeeping data (as in the laboratory measurements). Hence, times, in which the instrument is still adjusting to new surrounding conditions, are still part of the data used in the investigation. We have changed the wording to make this clearer.

*p17 line 9 - This sentence lacks a verb.*

The sentence has been corrected.

*p22 line 17 - If the authors choose to keep Section 5 as a discussion of new measurement strategy, it would be interesting to demonstrate the impact of LO shifts and/or elevation angle changes on simulated data.*

Since the authors decided to skip this section, and to only note the already shown considerations in an appendix, such work should be part of a stand-alone study to assess the impacts of measurement errors and possible changes in the measurement strategy on the retrieval output.

*p22 line 31 - "full-with-half-maximum" should read "full-width-half-maximum"*

The typo is corrected in the revised version of the document.

*p34 Figure 5 - A legend is needed here*

A legend with 6 different lines and their explanations would fill quite a large portion of the (already quite busy) plot. Hence, the meaning of the colours and line styles are explained in the figure caption.

*p40 Figure 11 and 12 - These figures are too small to differentiate the individual lines/methods.*

The two figures are now combined. In the upper panel (Figure 11 in the original manuscript) lines are clearly separated, displaying the offsets between individual methods. In the lower portion of the new figure (formerly Figure 12), the offset-correction is applied, and the plot demonstrates that this is a powerful correction, leading to very similar brightness temperatures being derived from all methods, so that the lines partly overlap.

*p44 Figure 15 (left panel) - It's impossible to distinguish the 58.363 GHz line from the 56.363 GHz line*

The 58.363 GHz lines are plotted in the background, and similar to lines of the other frequencies – hence, they are not well visible. We have added a sentence in the figure caption to point this out to the reader.

*p45 Figure 16 - The legend indicates lines for 6 altitudes are shown, but I can only see 4 on the left plot.*

At horizontal viewing angle, there is such a small difference between the lines representing altitudes at or below 8 km overlap in the left panel. We have added a sentence in the figure caption to point this out to the reader.

---

## Author Response (AR2)

AMT-2019-455

**Measurement Characteristics of an airborne Microwave Temperature Profiler (MTP)**

by Mareike Heckl et al.

Reply to the referee #1 comments

We would like to thank the referee for again thoroughly reviewing our manuscript and for the helpful advice provided in the comments, below, which we believe helped further improving the manuscript. In the following we give our answers regarding the points made by the reviewer. The statements, comments and suggested corrections raised by the referee are printed in black italics and our comments are presented in blue. We tried to consider all of the raised points in the revised manuscript in an adequate manner.

Answers to referee #1

*The manuscript has improved significantly, however, there are some flaws that need to be corrected. As a major issue a clear definition and subsequent wording of the terms how the measurement uncertainty characterized in terms of systematic and random components needs to be made (see also comments below). In this respect it might also be important to mention the uncertainty characteristics of the in-situ HALO-TS measurement which is given as 0.5 K. However, the standard deviation between HALO-TS and TB is ≤ 0.38 K which implies – together with the random noise characteristics of TB (around 0.3 K) that the precision (noise) of HALO-TS is much lower than that.*

We agree that the revised manuscript has improved. In the newly revised version, we have re-worded the text where necessary, according to the comments below, and making sure that the definition of noise characterisation is understandable and clear throughout the manuscript.
Concerning the noise characterisation of HALO-TS, the authors have cited available literature, which states the overall-error of HALO-TS (including accuracy). We support the observation that the precision of HALO-TS is better than that, (also implied by the overall accuracy of the sensors used to measure HALO-TS; see Ungermann et al., 2015, and supported by the observed RMS difference to the mean value of the 13s interval used in our study; see Table 7), however, we want to refrain from stating this in our manuscript without further investigation, and keep the cited value of 0.5 K overall uncertainty.

*The term "noise figure" is used frequently in the text at several instances when a general characterization of the noise is meant. However, the term "noise figure" is a typical specification for microwave elements (mixers, amplifiers) expressing the noise factor between in and output in decibels (given in dB). Therefore, please use this term with care. Most of the time it will be sufficient to just remove the word "figure" or replace it with "characterisation".*

We have followed the reviewer's suggestion to remove the word "figure" in most cases, and replacing it with "characterisation" in some instances. The changes can be seen in the marked-up document.

*Line 22. Another important application of HAMP is the liquid water path (see Jacob et al., AMT, 2020). In fact later on it would be good to mention that also non-resonant interaction of microwave radiation with atmospheric hydrometeors influences TB.*

The ability of HAMP to measure the LWP was added in the sentence with a reference to the paper by Jacob et al., 2019: https://doi.org/10.5194/amt-12-3237-2019.
Concerning the interaction with hydrometeors: We acknowledge the fact that hydrometeors can influence the measured TB, for example in case of aircraft icing. However, such situations are constantly avoided in flight. We also want to remind the referee that HALO is a high-altitude aircraft, with flight altitudes mostly above the tropopause, where water vapour concentrations are low. Still, the influence of hydrometeors is certainly important in the set-up of the retrieval algorithm, and should be discussed in that regard.

*P2,l2: Shorten to "Based on these mesoscale temperature fluctuation analyses, a number of modelling studies aimed at improving the understanding and numerical description of atmospheric gravity waves"*

We have made the suggested change.

*P3l15: I would always been careful to make such a general statement as:*
*"For the first time, this study presents all relevant instrument characteristics of the HALO-MTP instrument." Be aware that there might be always surprises ahead of the road. For example, I think that there is currently no chance to specify the absolute accuracy of the MTP as there is no absolute truth available – HALO TS is only an approximation as the authors discuss later on. Furthermore, in terms of spectral analysis investigations the current investigation is not all-encompassing. The Allan variance could still provide additional information especially in terms of the maximum time between two subsequent calibrations.*

The sentence was changed to: "For the first time, this study presents a thorough investigation of relevant instrument characteristics of the HALO-MTP instrument". We appreciate the fact that advances in science might lead to the insight that further characteristics have to be tested in order to fully understand MTP measurements – insights that might not be known today.
We do expect that the given uncertainty of MTP measurements provides a solid basis for confident and correct interpretation of MTP measurements (future and existing time series).

Concerning the Allen variance, we have pointed out, during the first review phase, that calibration measurements are already made as frequently as possible; i.e. during every single measurement cycle, corresponding to one calibration measurement every 13-14s, using standard measurement settings. As discussed previously, there is good reason to

assume that this is already the optimum of calibration measurements given the necessary trade-off between the time it takes to record a measurement cycle (i.e. best-possible horizontal resolution of measurements) and best-possible knowledge of the instrument state. A quick-look of an Allen Variance analysis from the laboratory measurements indicates that the current settings fall within the ideal timing of calibration measurements.

*P4l12: Please correct to "thermal radiation mainly emitted by oxygen .." Though the impact is low some influence of N2 and water vapor continuum occurrs especially when flying low. Similarly, hydrometeors can contribute to the signal.*

We agree with the reviewer's comment, and have made the suggested change. We would like to mention that basic constituents mentioned by reviewer one are included in the radiative transfer calculations presented in the appendix.
A few lines later we have extended the sentence: "The recorded TBs have to be converted to absolute temperature profiles by using a retrieval algorithm that utilises forward radiative transfer calculations. " to: "The recorded TBs have to be converted to absolute temperature profiles by using a retrieval algorithm that utilises forward radiative transfer calculations, *in which all possible impacts on the measured radiance (e.g. emission by other trace gases such as water vapour or nitrous oxide, or hydrometeors,…) have to be considered.*"

*P4, Eq. 2.1: Please mention that in this case the measured radiance I (usual notation) is equal to the Brightness B of a blackbody described by Plank's law. Also note that the omission of the higher order terms in the Rayleigh-Jeans approximation can be more than 0.5 K at your frequencies, check table 1 in Liu et al., 2008 https://doi.org/10.1016/j.jqsrt.2008.03.001*

The information was added in the manuscript text, directly after the equation. The assumption that the measured signal has a linear relationship to the source temperature has been tested in the lab and it was shown that a calibration based on this assumption leads to reliable results in determining the brightness temperature. The point raised about using the Rayleigh-Jeans approximation rather than Planck's law again relates to radiative transfer calculations, which are part of the retrieval. Hence, the uncertainty in the Rayleigh-Jeans approximation and the corresponding error in the converted brightness temperature will not be explicitly discussed in this study.

*P4, l26: The horn is no receiver. Please make also the connection to the antenna response function: "..a horn antenna guides the incoming atmospheric radiation and determines the spatial response function". What about the rotating mirror – is it just flat or does it focus the antenna beam?*

We have corrected the sentence in the manuscript following the reviewer's suggestion, adding a few words concerning the mirror: "a horn antenna guides the incoming atmospheric radiation and, together with the hyperbolic shape of the rotating mirror, determines the spatial response function": The mirror is not flat, but is used to focus the antenna beam. On page 8 of the manuscript we already note that the antenna function is mainly defined by the shape of the rotating mirror. Also, in Table 1 we already point out that the mirror was explicitly designed to provide a beam width of 7.5 degrees of the antenna function. Here, it was now added, that the mirror has a hyperbolic shape.

*P4, l28. Why don't you use the common term interim frequency (IF) instead of base-band? Further you could also provide the information that this is around 100 MHz which makes it much clearer for the reader in the following. Suggest the following text:*
*".. the incoming signal is converted to the interim frequency (IF) around 100 MHz. Both difference frequencies below and above the LO frequency are down-converted to the IF in the double side band receiver. Low pass filtering supresses any incoming radiation outside the IF bandwidth of 200 MHz such that the symmetric spectrum around the current LO frequency is measured with only a minor gap of approx. 20 MHz at the LO frequency. The IF signal is converted to a voltage.." In this way the information on Page 5 last line is not necessary. I think this information fits much better here – or is the IF bandwidth different for the wing-canister instrument than for other MTP?*

We have made the suggested changes with the according information.
The HALO-MTP does indeed have slightly different operational settings than older versions of the MTP, which would be useful for the reader to know. The HALO-MTP does indeed operate at base-band, while older versions operated at 320 MHz (Mahoney, Denning, and Lim et al., priv. comm.).

*P5, l2: "The PHYSICAL temperatures.."*

We have added the word "physical" as suggested.

*P5, l3: Also mention the amplifier which becomes important later on.*

The whole radiometer plate, which includes the mixer and amplifiers is temperature-controlled. Hence, the amplifiers are also listed now.

*P6, l2: "..doubled twice, POTENTIALLY allowing …Here only the standard set… is used." In the beginning I was confused here so this might also help others.*

We have made the suggested changes in the manuscript.

*P6l25: I find "half-hemisphere" =quarter sphere rather confusing. It is not necessary to mention it but rather it is important to say what "Both functions" in the next sentence means.*

We have changed the sentences to: "… to the different directions it is pointing towards. Both, the filter functions, as well as the antenna diagram, have been measured …".

*P6/5: I don't think that the logic for the second part of the sentence is correct "despite the fundamental assumption in MTP calibration that this relation is always linear ". I suggest to change to "Amplifiers might change their characteristics and thus the relation between the recorded signal and the source temperature, i.e. the calibration parameters, change." You still assume linearity. For nonlinearity either higher order terms or an exponential relationship would be need to be considered in the calibration equation and determined by introducing a third reference point.*

We agree that the suggested sentence is clearer, and have made the according change in the manuscript.

*P7,l4: noise diode signal strength*

"strength" was added after "noise diode signal".

*P7,l24: "The gap in the centre is due to the use of a double-side-band receiver as explained in Sec?"*

The suggested change has been made, including a reference to Section 2.

*P8,l3: Anntenna pattern: "It is actually mainly defined by the shape of the rotating mirror at the front of the instrument." The text did not provide any information on the shape and I can not see something in Fig. 1. If it is a flat mirror then only the horn is important. If it is used for beam forming, e.g. an offset parabolic mirror, it is important to mention that? Has the antenna beamwidth not been calculated using gaussian beam optics?*

In Table 1 we explicitly mentioned that the rotatable mirror is used to define the beam shape. We have now added that it is hyperbolic. The beam shape was measured in the laboratory, as described in the manuscript, and the parameters stated in the text are derived from that measurement, confirming the numbers provided by the manufacturer.

*P9,l11: "itself IS monitored"*

We have made the suggested change.

*P9,l11: Over this large temperature range it is difficult to see by eye if a linear relation ship fits the data better than a non linear/exponential one? This comment is maybe a bit picky but one could easily check the significance of the statement. What I further notice is that the slope for the dark blue load seams to be lower than the one of the light blue load for the lower two channels and vice versa for the upper channel. As this is shown in counts rather than in brightness temperatures it is difficult to judge if this is significant. Why don't the authors use a simple/standard calibration that would help the reader to judge the impact – this could just be plotted on the second y-axis for illustration – of course an offset needs to be introduced to separate the channels? This comment also holds for the following two figures as nobody is interested in counts,*

This comment relates to figure 5, which illustrates the direct linear relationship between the measured signal (given in counts) and the source temperature (ambient targets at different positions within the cold chamber). A conversion to brightness temperatures already uses a calibration calculation based on some standard values, which – as is shown later in the manuscript – clearly depend on the current cold-chamber temperature, and corresponding instrument state. Hence a conversion to brightness temperatures may lead to a wrong interpretation of data. This, along with the other findings that calibration coefficients differ for different frequencies, has led to the decision to plot the raw signal in counts, and we would refrain from changing the figures in the manuscript.
When performing quadratic or even cubed fits to the data, the parameters clearly indicate that only the linear term contributes significantly (fit coefficients at least three orders of magnitude larger than the quadratic or cubed term). We have added a sentence in the manuscript reflecting this.
On the comment concerning the differences between the dark and the light blue lines: Please note that due to the limited space available, one of the targets is placed closer to the ventilation of the chamber than the other, so the two ambient targets do not have the exact same temperature!

*P9,l30 – same comment relate counts to Tb*

The sentence was changed to: " … can be characterised by a Gaussian distribution with a 30 standard deviation of approximately 6 cnts (approx. 0.25 K ) and the mean at 0 cnts.

*P10, l11: In addition to the term "noise figure" the use of "long-term stability" is unclear. Suggest: "This is strong evidence that the HALO-MTP noise characteristics do not change between flights, and the laboratory assessment can be used to characterize measurements of different campaigns and serve as information for the retrieval development."*

The sentence was changed to: "This is strong evidence that the HALO-MTP noise characteristics do not change between flights, and the laboratory assessment can be used to determine overall instrument-health and comparability of measurements in between campaigns, also serving as information for the retrieval development."

*P12, l20: In fact if you are flying high (thus having an optical thin atmosphere above you) and assume horizontal homogeneity you could do the tipping curve calibration from the different elevation angle measurements pointing upwards (probing different opacities). For sure, I don't expect that you do that but it is not the instrument design limiting you.*

While according to Han and Westwater, 2000 the antenna beam width is a major source of uncertainty in this calibration approach, we agree that there are other limiting factors, too.

The sentence has been changed to: Other calibration methods, such as the tipping curve calibration (e.g. Küchler et al., 2016, or Han and Westwater, 2000) are not available for the DLR MTP due to the given instrument design (mainly antenna beam width), potentially fast-changing atmospheric conditions, due to the moving platform, influencing radiative transfer calculations needed in this approach, as well as the need for an efficient measurement strategy.

The reference to Han and Westwater, 2000 was added in the reference list.

*P14 l7: Please again relate counts to TB*

We added that 20 counts correspond to approximately 0.83 K (using the reference calibration parameters from Table 7).

*P15, l18 – suggest to say absolute accuracy*

We have made the suggested change.

*P15, l29: "The accuracy of the temperatures " – be more specific. Which T, which type of accuracy?*

We have added the word "brightness" to make clear, that brightness temperatures are meant.

*P16, l17: Eq. 4.5 give the noise equivalent temperature NeDT not the measurement uncertainty. This is not the variance (rather square root of variance) of the measurement noise but the radiometric resolution (see Eq. 6.13 by Woodhouse, 2017) and can be estimated as standard deviation from time series, e.g. from Fig.6 I would estimate something like 0.3 K. This already roughly corresponds to the values in Fig. 11. Note, the effect of the*

*passband is negligible compared to other deviations from an ideal radiometer and should not be mentioned.*

Equation 6.52 in Ulaby et al., 1981, which is the one stated in the Manuscript, is derived from the definition of the variance. We have changed the equation in the manuscript to look the same as in Ulaby et al., 1981, and used the term $\Delta T_{theo}$ in the text for continuity. As explained in the end of the paragraph, this theoretical equation does not consider any gain variations.

The gain variation is still part of the calculation of the standard deviation (Figure 7), since we only subtracted the linear fit (drift of the instrument), not the running average (blue line in Figure 7). Only that would have included the gain variations, and would have led to a similar value as the theoretical one. However, that value would not be representative of the real flight situations, where changes of the atmospheric temperature are frequently observed, and must remain detectable in the MTP data.

*Maybe it is good to clarify that in its simplest concept measurement uncertainty includes systematic errors (bias, absolute accuracy) and random errors (noise, precision).*

We have added a sentence in the beginning of the section stating: "The overall uncertainty of the MTP measurements includes both systematic errors (e.g. the bias, described above), and random errors (e.g. noise). In case of the HALO MTP, the former can be related to HALO TS, as shown above, the latter mainly influences measurement precision, and influences the ability of the instrument to pick up atmospheric temperature fluctuations. In the literature (e.g. Ulaby et al., 1981), …"

*P18, l14: can you clarify whether this is random or systematic?*

The deviation is mainly a systematic error (the biggest source being the aircraft pitch, which is now stated in the manuscript); however, it is dependent on the flight conditions, as the offset changes with speed and altitude of the aircraft. Possible vibrations may add a random error contribution. Since there is not enough data to make statistically significant evaluations of the offset, we cannot say more than is already stated in the manuscript.

*P18, l26: It is likely only relevant if hydrometeors or WV variations occur,,*

This comment refers to the discussion of synthesizer errors. We agree that those are not a large error source, as discussed in the manuscript. It is clear that the frequency-dependent influence of water vapour would only be noticeable, if strong variations in the frequency would occur. The argument stated in the text is still valid: the synthesizer error must be large enough to cut out a portion of the Oxygen absorption line to make the changed influence of water vapour absorption noticeable. In that case, however, probably the whole radiative transfer calculation must be adapted anyway, and the health of the instrument should be checked.

*P20, l7: please transfer to Tb and discus in respect to the later results and the term precision. It is especially important here as you want to give an idea which amplitude of gravity waves (be clear on that) can be detected.*

6 counts correspond to slightly less than 0.25 K, which was added in the text. We also added a half-sentence at the end of the paragraph, so that the last sentence now reads:
" […] the presented characterisation of the HALO-MTP noise allows the identification of significant atmospheric signals in MTP measurement time series, as long as the amplitude of

the atmospheric signal is larger than the measurement precision (i.e. significantly larger than 0.38 K), and wave-like signals (e.g. caused by gravity waves) can be clearly separated from the noise-induced structures, caused by the auto-correlation of MTP measurements.".

*P20l20: "To achieve this accuracy, the necessity of an offset-correction relative to HALO TS" Here you talked about precision not about the bias. Just delete the first part "To achieve this accuracy"*

We have made the suggested change in the manuscript.

*P20, l30. "clearly dominated by the contribution from measurement noise." Is this really true if I look at Fig. 10.*

In the manuscript, the following sentence explains that other potential sources of uncertainty for brightness temperatures (i.e. the retrieval input) do not have as much influence on the overall uncertainty than other potential sources of uncertainty. Potential drifts seen in Figure 10 (which depicts data recorded during a measurement flight, not in the laboratory), are most likely caused by changing atmospheric conditions. We remind the reviewer that the offset-correction only uses one single value; the leg-mean HALO-TS temperature. It is important that atmospheric temperature changes occurring during a flight leg remain detectable, as seen in Figure 10. We have added this in the manuscript text (both, in Section 4, as well as in the Summary).

*P21, l12: "all necessary instrument parameters" I wouldn' be so confident!*

The sentence was changed to: "Overall, this study summarises the investigation of instrument parameters and characteristics, necessary to accurately analyse…"
It was clearly stated in our study that there are limitations to what can be characterised in this case, and we have performed those characterisations that are necessary and helpful to understand the recorded data (i.e. being aware of uncertainties in data time series, and knowing the abilities of the instrument). Thus, we are confident, that all necessary information is provided in this study to allow for correct data interpretation and evaluation. At the same time, we do acknowledge the reviewer's comment from the beginning; that – as this is the nature of scientific progress – the future might show that there might be some detail that needs to be considered, of which we cannot be aware of at this point.

[revised manuscript text omitted]